# A genome-wide CRISPR screen in *Anopheles* mosquito cells identifies fitness and immune cell function-related genes

Enzo Mameli [1,4], George-Rafael Samantsidis [2,4], Raghuvir Viswanatha [1], Hyeogsun Kwon[2], David R. Hall [2], Matthew Butnaru[1], Yanhui Hu [1], Stephanie E. Mohr [1], Norbert Perrimon [1,3,5] ✉ & Ryan C. Smith [2,5] ✉

*Anopheles* mosquitoes are the sole vector of malaria, the most burdensome vector-borne disease worldwide. At present, strategies for reducing mosquito populations or limiting their ability to transmit disease show the most promise for disease control. Therefore, improving our understanding of mosquito biology and immune function may aid new approaches to limit malaria transmission. Here, we perform genome-wide CRISPR screens in *Anopheles* mosquito cells to identify genes required for fitness and that confer resistance to clodronate liposomes, which are used to ablate immune cells. The cellular fitness screen identifies 1280 fitness-related genes (393 at highest confidence) that are highly enriched for roles in fundamental cell processes. The clodronate screen identifies resistance factors that impair clodronate liposome function. For the latter, we confirm roles in liposome uptake and processing through in vivo validation in *Anopheles gambiae* that provide new mechanistic detail of phagolysosome formation and clodronate liposome processing. Altogether, we present a genome-wide CRISPR knockout platform in a major malaria vector and identify genes important for fitness and immune-related processes.

Mosquitoes are essential vectors for the transmission of a variety of bacterial, viral, and parasitic pathogens that cause significant socio-economic burden and mortality across the globe[1]. Among mosquito-borne diseases, malaria causes more than 200 million clinical cases and 600,000 deaths every year[2], and is transmitted exclusively through the bite of an *Anopheles* mosquito. Given their public health importance, mosquitoes have become an emerging model system to examine aspects of development[3], blood-feeding physiology[4], vector-pathogen interactions[5], and gene-drive technologies[6]; each with the ultimate goal of developing approaches to reduce transmission of mosquito-borne diseases.

Significant progress in our understanding of mosquitoes has been made using reverse genetic approaches utilizing tools such as RNAi[7], transgenesis[8,9], and site-directed mutagenesis[10–14]. The development of forward-genetic technologies would make it possible to associate genes with phenotypes in an unbiased manner, thereby enabling the discovery of both conserved and mosquito-specific gene functions. New advances in CRISPR technology have made it easier to perform genetic studies in mosquitoes and other non-model species[15]. However, what has remained lacking is an efficient system for genome-wide forward genetic screening using CRISPR or other similar technologies.

[1]Department of Genetics, Blavatnik Institute, Harvard Medical School, Boston, MA, USA. [2]Department of Plant Pathology, Entomology and Microbiology, Iowa State University, Ames, IA, USA. [3]HHMI, Harvard Medical School, Boston, MA, USA. [4]These authors contributed equally: Enzo Mameli, George-Rafael Samantsidis. [5]These authors jointly supervised this work: Norbert Perrimon, Ryan C. Smith. ✉e-mail: perrimon@genetics.med.harvard.edu; smithr@iastate.edu

To address this, we recently developed a platform for pooled-format CRISPR screening in mosquito cells based on similar platforms we developed for *Drosophila* cells[16,17]. For this approach, we use recombination mediated cassette exchange (RMCE) to integrate single guide RNAs (sgRNAs) into the genome, making it possible to later associate screen assay phenotypes with genotypes. Application of this approach in *Drosophila* cells has resulted in the identification of genes required for fitness[16,18], a novel transporter for the insect hormone ecdysone[19], and receptors of bacterial toxins[20]. To extend this approach to *Anopheles*, we first engineered the *Anopheles* Sua-5B cell line with attP sites for RMCE and stable expression of Cas9 (i.e., a 'screen-ready' cell line); identified pol III promoters for sgRNA expression in *Anopheles* cells; and developed an approach to sgRNA design for screens in *Anopheles* Sua-5B cells. We next demonstrated the utility of this approach by introducing a library of 3,487 sgRNAs into screen-ready Sua-5B cells and screening for cells resistant to rapamycin, ecdysone, or trametinib[21]. As expected, we were able to precisely and efficiently identify the *Anopheles* orthologs of the targets of these well-known treatments[21], opening the doors

to the application of large-scale forward-genetic screening in *Anopheles* cells.

One of our goals in developing the *Anopheles* screening platform was to further contribute to understanding mosquito immune responses and cellular immune function. Mosquito immune cells, known as hemocytes, are essential components of the innate immune system[22] and influence mosquito vector competence to both arbovirus[23,24] and malaria parasite infection[25–29]. With few genetic resources available to study mosquito hemocytes in vivo, we recently adapted the use of clodronate liposomes, used in mammalian systems for macrophage depletion[30,31], to chemically ablate macrophage-like immune cell populations (granulocytes) in arthropods[28,32,33]. This methodology has been instrumental in uncovering how mosquito granulocyte populations contribute to host survival and pathogen infection outcomes[24,28,32]. However, despite their widespread use in vertebrate and invertebrate systems, the mechanisms by which clodronate liposomes enter and promote targeted cell ablation remain unclear. We therefore reasoned that the application of a genome-wide CRISPR screen might identify factors required for clodronate

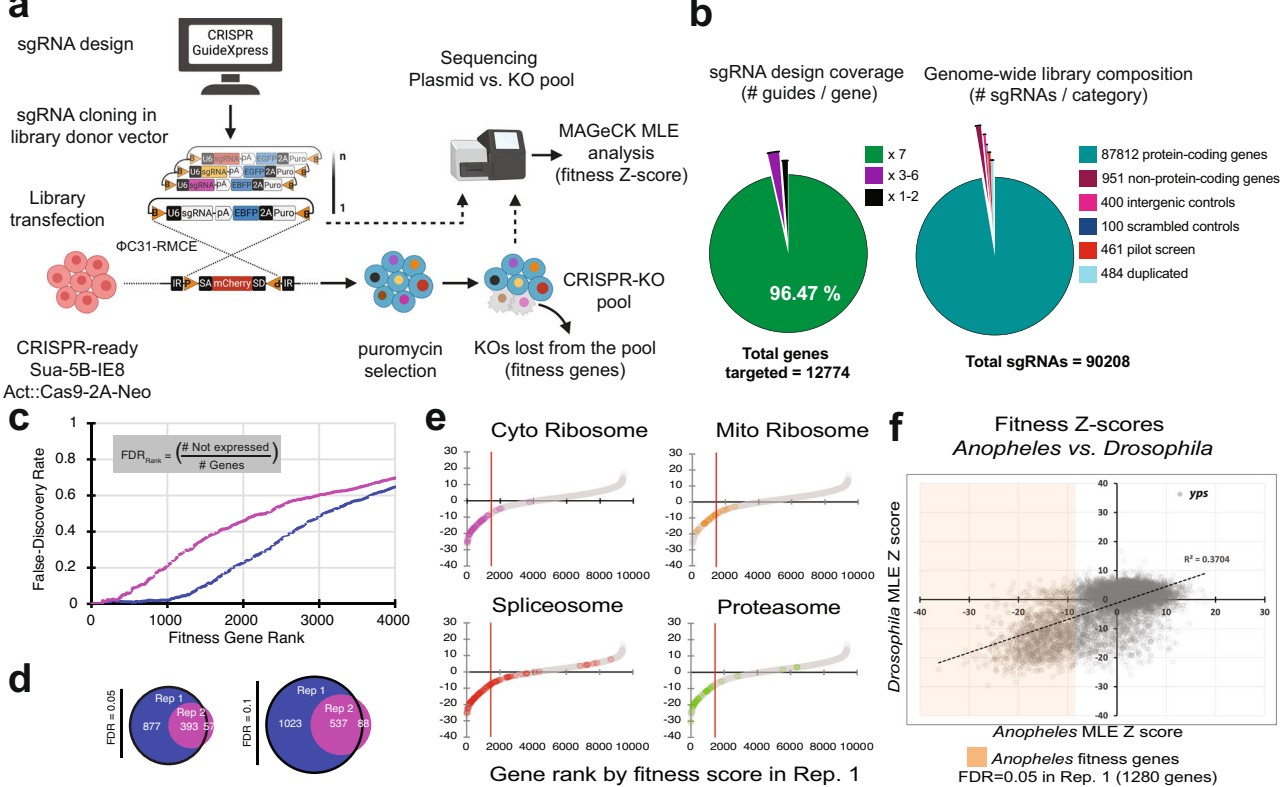

**Fig. 1 | Genome-wide CRISPR knockout screen reveals genes required for fitness in *Anopheles* Sua-5B cells. a** Schematic for the fitness gene CRISPR screen. CRISPR GuideXpress was used to design a whole-genome sgRNA library targeting protein-coding and non-coding *Anopheles gambiae* genes. The library was cloned into the pLib6.4B_695 vector and delivered to Sua-5B-IE8-Act::Cas9-2A-Neo via ΦC31 recombination-mediated cassette exchange to yield a pool of knockout cells. During outgrowth, cells that received sgRNAs targeting fitness genes will "drop out" of the KO pool. The relative abundance of each sgRNA in the KO outgrowth pool of cells was compared to the plasmid library by NGS followed by MAGeCK MLE analysis. **b** Genome-wide library coverage. The starting library design includes 90,208 sgRNAs (88,763 unique sgRNAs) targeting 93% of *Anopheles* genes, with 7 sgRNAs per gene coverage for ~96% of these genes. Following the screen assay, data were analyzed using MAGeCK MLE. Shown in (**c**) is the cumulative distribution of false discovery of fitness genes (genes with low expression, TPM < 1) for each replicate. Replicate 1 identified a larger number of fitness genes at a low false

discovery cut-off (1280 genes at FDR = 0.05). The distribution of genes by category is shown in (**e**). All data points within the Z-score whole genome distribution for replicate 1 are displayed (gray or colored circles); colors highlight genes annotated with the indicated Gene Ontology term (Cytoplasmic Ribosome KEGG:aga030008; Mitochondrial Ribosome GO:0098798,0005763; Spliceosome KEGG:aga03040; Proteasome KEGG:aga03050). The red line intercept of the *x* axis represents Z-score threshold at FDR = 0.05. **f** Comparison of fitness genes identified in *Anopheles* and *Drosophila*. To compare gene lists between the two species, *Anopheles* genes were mapped to corresponding *Drosophila* orthologs, then scores from *Anopheles* replicate 1 or *Drosophila* knockout screens were plotted. Colored box within the plot highlights *Anopheles* fitness genes with Z-scores within FDR = 0.05 in replicate 1; fitted linear trendline and R² squared value are displayed to highlight the correlation trend between datasets; *yps* datapoint was darkened to enhance its visibility. Image in **a** was created using BioRender. Mameli, E. (2025) https://BioRender.com/rkhue0x. Source data are provided as a Source Data file.

liposome-mediated cell ablation, enhancing our understanding of clodronate liposomes as a research tool and the potential immune-related mechanisms shared between liposome and pathogen uptake.

Herein, we perform two genome-wide CRISPR screens in *Anopheles* cells, one identifying fitness-related genes and the other identifying genes involved in the uptake and processing of clodronate liposomes. Our results demonstrate the potential of forward-genetic screens in mosquito cell lines to advance our understanding of cellular immune function and contribute to the development of new mosquito control strategies.

## Results

### Genome-wide CRISPR knockout screen to identify fitness genes in *Anopheles*

To extend the pooled screen approach genome-wide, we first cloned a library of 89,711 unique sgRNAs targeting 93% of *Anopheles* genes, with ~96% of these genes targeted by 7 sgRNAs per gene, based on our previously reported sgRNA design resource for this species[21] (Fig. 1a, b). The *Anopheles* gene-targeting sgRNAs were supplemented with positive and negative control sgRNAs, resulting in a total library of 90,208 sgRNAs (Supplementary Table 1 and Supplementary Data 1). We then introduced the library into CRISPR 'screen-ready' (attP + , Cas9 + ) *Anopheles* Sua-5B cells[21] in the presence of ΦC31 integrase to generate a pool of knockout (KO) cells (Fig. 1a). Our first goal for genome-wide screening was to use a 'dropout' assay (negative selection assay) to identify genes for which knockout results in decreased fitness, growth arrest, and/or cell death (hereafter, "fitness genes"). For each of two biological replicates, at 8 weeks of outgrowth of the KO cell pool, we compared the relative abundance of each sgRNA in the outgrowth pool to the distribution of sgRNAs in the starting plasmid library by next generation sequencing (NGS) followed by MAGeCK MLE analysis[34] (Supplementary Data 2). Based on the NGS results, we conclude that sgRNA delivery was higher for replicate 1 than replicate 2, likely due to variation in transfection efficiency. Using the relationship between gene expression and Z-score rank, for replicate 1 we identified 1280 putative fitness genes with 95% confidence (FDR = 0.05) (Fig. 1c and Supplementary Data 2). Of these, 393 were also identified in replicate 2 at 95% confidence (Fig. 1c, d and Supplementary Data 2). These include *Rho1* (AGAP005160), an experimentally validated fitness gene[21]. Furthermore, we compared beta and Z-score values between replicates and found that the beta scores are more closely correlated than the Z-scores ($R^2 = 0.33$ for beta scores vs. $R^2 = 0.25$ for Z-scores; Supplementary Fig. 1). Beta scores estimate the raw effect size of a gene knockout on the phenotype, whereas Z-scores normalize these effect sizes by their variability to indicate how statistically significant the observed effect is relative to background noise[35].

We next analyzed the larger set of 1280 putative fitness genes identified in replicate 1. The majority of these genes are annotated as components of the cytoplasmic or mitochondrial ribosome, spliceosome, or proteasome, resulting in negative Z-scores (Fig. 1e). Moreover, we observed a relationship between gene fitness and transcript abundance, where fitness genes are markedly distributed towards genes with the highest expression across the transcriptome (Supplementary Data 3), as expected for essential cellular functions such as translation. Surprisingly, 27.6% (353/1280) of identified fitness genes were classified as "non-expressed" in our transcriptomic dataset ($\log_{10}[\text{TPM} + 0.01] < 1$), suggesting that bulk RNA-seq may not capture low-level, transient, or cell-state-specific expression patterns relevant for fitness. We next identified *Drosophila* orthologs of the *Anopheles* replicate 1 fitness genes using DIOPT[36] (v9.0), then performed gene set enrichment analysis (GSEA) using PANGEA[37] (Supplementary Fig. 2 and Supplementary Data 4). Generic gene ontology (GO) terms for biological process[38] were enriched for fundamental cellular processes using GSEA and when the curated Gene List Annotation for *Drosophila* (GLAD) resource[39] was used as a reference, we found significant enrichment of gene groups

corresponding to fundamental components of the ribosome, proteasome, and spliceosome (Supplementary Fig. 2). Similar GSEA using PANGEA was used to examine phenotypes associated with classical mutations as annotated by FlyBase[40], with "cell lethal" the top-enriched phenotype and producing other enriched phenotypes such as "decreased occurrence of cell division," "abnormal cell cycle," and "abnormal cell size" (Supplementary Fig. 2). Supporting these enriched phenotypes, we identified a ribonucleoprotein complex component[41], *ypsilon schachtel* (*yps*) (AGAP006108; FBgn0222959), which appears to limit cell growth in both *Anopheles* (Fig. 1f, upper-right quadrant) and *Drosophila* cells[16].

The Sua-5B cell line has been characterized as "hemocyte-like"[42] and thus, the fitness genes identified may most closely resemble genes required for hemocyte fitness. To investigate this line of reasoning, we next asked if Sua-5B RNA-seq data align with bulk RNA-seq and single-cell RNA-seq expression profiles for *An. gambiae* hemocytes[43,44]. Using randomization and hypergeometric statistical tests, we found a significant concordance between genes expressed in Sua-5B cells and hemocytes, including subsets of genes assigned to specific hemocyte clusters (Supplementary Data 5). We next compared genes expressed in hemocytes with the set of fitness genes identified in the screen[42]. The overlap includes *serpent* (*srp*; AGAP002238), an ortholog of the GATA transcription factor involved in hematopoiesis in *Drosophila*[45,46] (FBgn0003507) (Supplementary Data 5). We then used RNAi to silence *srp* in adult *An. gambiae* females, which resulted in reduced hemocyte numbers and an increased intensity of malaria parasite infection (Supplementary Fig. 3). This supports that *An. gambiae* srp has similar roles in mosquito hematopoiesis and immune function.

### Comparisons of fitness genes in *Anopheles*, *Drosophila*, and human cells

We next compared putative fitness genes identified in replicate 1 of this screen with genes identified using similar approaches in *Drosophila*[18]. To do this, we mapped the 1280 *Anopheles* genes to *Drosophila* orthologs using DIOPT[36] (v 9.0) and filtered the results based on the DIOPT score. For each orthologous gene pair, we graphed the corresponding Z scores, observing a high degree of correlation between species (Fig. 1f, lower-left quadrant). We next asked how many of the *Anopheles* genes that score in replicate 1 as fitness genes (FDR = 0.05) and have *Drosophila* orthologs overlap with *Drosophila* fitness genes defined in a CRISPR knockout screen in *Drosophila* S2R+ cells[18]. The 1280 mosquito genes mapped to 1219 *Drosophila* genes, and 88% of these (1073/1219) were identified as fitness genes in the *Drosophila* cell screen (Supplementary Data 3).

We then mapped genes identified in the *Anopheles* fitness gene list to human orthologs, and asked how many are included in a list of 684 "human core essential genes" compiled from 17 human cell knockout screens[47]. The 1280 *Anopheles* fitness genes mapped to 1185 human orthologs and of these, 34% (398/1185) were among the human core essential genes (Supplementary Data 3), suggesting that the roles of these 398 genes are conserved among distantly related metazoan cells.

### Genome-wide CRISPR screen for resistance to clodronate treatment

Clodronate liposomes can be used to probe cellular immune function in arthropods[28,32,33], yet even in mammalian cells where their use is more established[30,31], we lack an understanding of how they promote cell ablation. Therefore, we reasoned that a similar genome-wide CRISPR screen to examine clodronate liposome-mediated cell ablation in *Anopheles* Sua-5B cells could reveal important factors relevant to clodronate liposome function in mosquito immune cells. To do this, we first tested the effects of treatment on "screen-ready" Sua-5B cells with a range of concentrations for clodronate liposomes or control (empty) liposomes to determine an appropriate concentration for selection-based screens (Fig. 2a). We found that the $IC_{50}$ of the

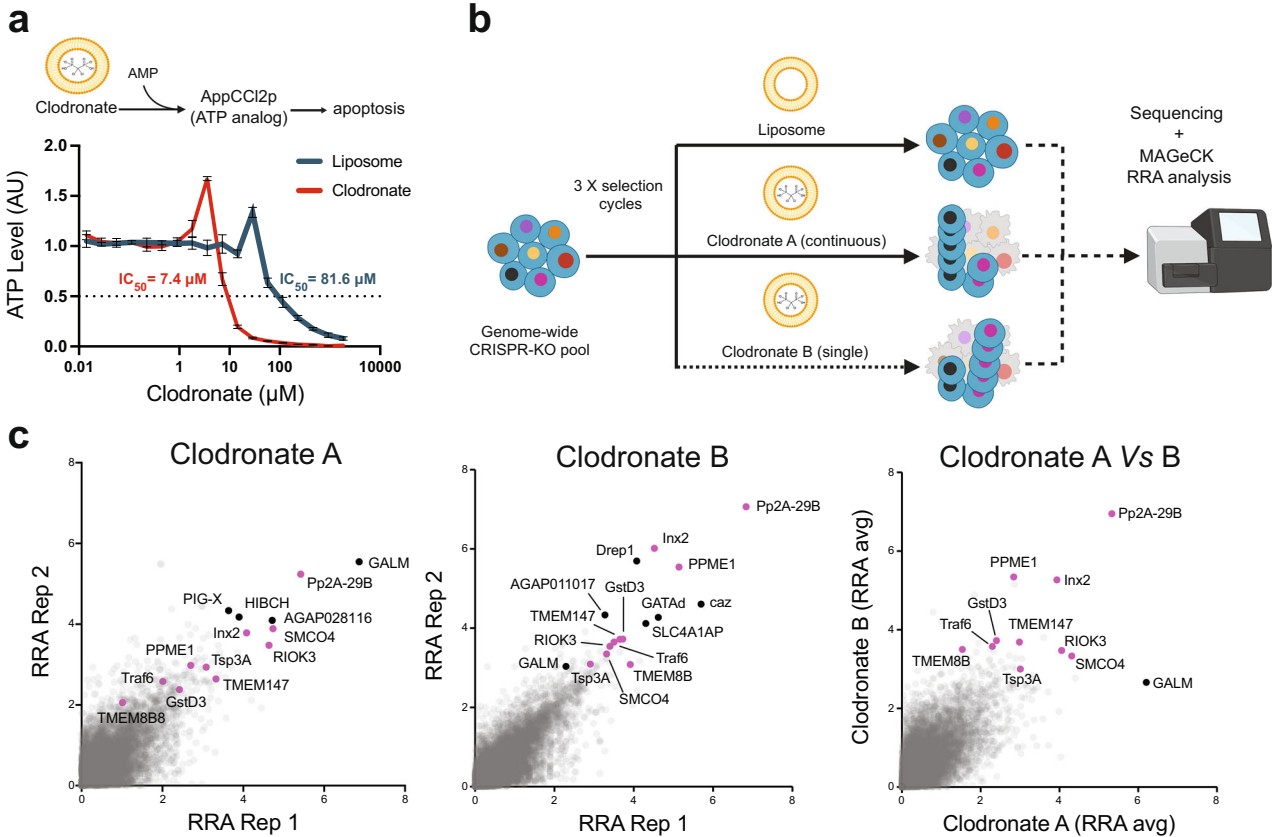

**Fig. 2 | Genome-wide CRISPR knockout screen reveals genes for which knockout confers resistance to clodronate liposome uptake and/or induced cell death in *Anopheles* cells. a** Clodronate liposomes induce cell death after cellular uptake by releasing clodronate, which is enzymatically converted to adenosine 5′-β-γ-dichloromethylene triphosphate (AppCCl2p), an ATP analog that can induce apoptosis. Assay of total ATP levels reveals higher lethality in the *Anopheles* Sua-5B-IE8-Act::Cas9-2A-Neo cells treated with clodronate (red) as compared to a liposome control (blue; 11-fold difference in relative IC50 values). Error bars represent the standard deviation of the mean calculated from three technical replicates. IC50 values were calculated using a four-parameter logistic regression model "Quest Graph™ IC50 Calculator." AAT Bioquest, Inc., https://www.aatbio.com/tools/ic50-calculator. **b** Schematic of a genome-wide positive selection CRISPR knockout screen for clodronate resistance. A genome-wide CRISPR KO pool of Sua-5B-IE8-Act::Cas9-2A-Neo cells was left untreated or treated with 16 μM liposome (control) or 8 μM clodronate. All treatments were performed for three cycles and with continuous drug selection of the KO pool, except for the "Clodronate B" treatment group, for which cells were subjected to an initial treatment for 4 days and then allowed to recover in non-selective media at each treatment cycle. Genomic DNA from endpoint cell populations was used for PCR amplification of sgRNAs, followed by NGS and enrichment analysis using the MAGeCK robust rank aggregation (RRA) algorithm. **c** Scatter plots of RRA scores for two replicates, comparing clodronate treatments A and B to control liposome treatments (left and center panels) or comparing average RRA scores between the two treatments (right panel). Left and center panels: The 'hits' (positive results) are labeled with gene symbols. Hits shown in black represent the top eight genes by RRA rank. Hits in magenta are genes of interest selected for further analysis from among the top 50 hits from each screen. Right panel: only genes of interest are labeled, while the top hit gene of the clodronate A screen (in black) is shown for reference. Gene symbols shown are the symbols for orthologous human genes (symbols in all caps) or orthologous *Drosophila* genes. Images in a and b were created in part using BioRender. Mameli E. (2025) https://BioRender.com/0y863vx and assembled in Adobe Illustrator. Source data are provided as a Source Data file.

clodronate liposomes for Sua-5B cells was 7.4 μM, whereas the IC50 of control liposomes was >11-fold higher (81.6 μM; Fig. 2a). Using this information, we subjected a pooled library of Sua-5B KO cells to continuous selection with clodronate liposomes ("Clodronate A" group); to treatment for 4 days with clodronate liposomes followed by outgrowth in standard media ("Clodronate B" group); or to continuous treatment with control liposomes for a total of three cycles of treatment/outgrowth ("Liposome" group) (Fig. 2b). Following the last cycle of outgrowth, we used deep amplicon sequencing and MAGeCK analysis[34] to compare sgRNA abundance in each treatment group (Supplementary Data 1 and Supplementary Data 6).

To identify candidate genes involved in clodronate uptake and/or processing, we selected for genes enriched in either clodronate treatment group (Clodronate A or B) as compared with the liposome control (Fig. 2c). The top-scoring gene in the continuous treatment (Clodronate A) group is a predicted *Anopheles* ortholog of the mammalian *GALM* (AGAP008154), whereas the top-scoring gene in the Clodronate B group is a predicted ortholog of mammalian

*PPP2R1A* and *Pp2A-29B* in *Drosophila* (herein referred to as *Pp2A-29B*; AGAP009105). While some top-scoring genes were different between the two treatment groups, twelve genes scoring in the top 50 hits were in common (Fig. 2c): *Pp2A-29B*, *PPME1* (AGAP008336), *SMCO4* (AGAP003534), *RIOK3* (AGAP009993), *Inx2* (AGAP001488), *PAFAH1B2* (AGAP000939), *Tsp3A* (AGAP002257), *TMEM147* (AGAP008757), *FAM117B* (AGAP011572), *jbug* (AGAP007006), *caz* (AGAP001645), and AGAP011017. To reveal the genetic determinants of clodronate liposome uptake and induced toxicity, we performed similar GSEA analyses as for the fitness gene set (Supplementary Data 4) using *Drosophila* orthologs of the top scoring genes from the clodronate liposome treatment resistance screens. GSEA was performed with GO biological process terms from standard GO sets ("GO hierarchy" at PANGEA); GO subsets specifically curated for *Drosophila* by Flybase[40] and the Alliance for Genome Resources[48] (Slim2); and FlyBase Gene Groups[49]. Methyltransferases were commonly enriched across analyses, with genes associated with "nitrogen compound metabolic process"

enriched in the *Drosophila* GO subset analysis (Supplementary Fig. S4 and Supplementary Data 6).

## Optimization of clodronate liposome concentrations and timing of uptake in vivo

Previous in vivo studies using clodronate liposomes in *An. gambiae* were performed using a concentration of ~120 µM (1:5 dilution)[28,29,44], a concentration much higher than the ~8 µM used herein for the cell screens (Fig. 2a). To confirm that this lower concentration is able to promote cell ablation in vivo, we compared the efficiency of clodronate liposomes at the 1:5 dilution with a 1:50 dilution (~12 µM; comparable to the cell screens). Using the expression of *eater* and *Nimrod B2* as a proxy for mosquito immune cell (granulocyte) numbers as previously established[28,29,32,33,44], both the 1:5 and 1:50 clodronate liposome dilutions were able to promote similar reductions in *eater* and *Nimrod B2* (Supplementary Fig. 5), suggesting that both concentrations were equally effective in their ability to reduce mosquito immune cell populations in vivo. Moreover, the 1:50 clodronate liposome treatment effectively reduced granulocyte populations for more than 10 days (Supplementary Fig. 5).

Similarly, while previous studies have demonstrated the utility of clodronate liposomes to deplete immune cell populations in flies, mosquitoes, and ticks[28,29,32,33,44], the precise timing required for phagocyte depletion has not been previously examined. Therefore, we utilized fluorescent liposome particles (LP-DiO) to determine the temporal kinetics of liposome uptake and subsequent phagocyte depletion. Uptake of fluorescent LP-DiO particles peaked at 6 h post-injection (with ~37% of hemocytes LP-DiO⁺), before the percentage of LP-DiO⁺ cells began to decrease over time (Supplementary Fig. 6). To validate these findings in the context of clodronate-mediated phagocyte depletion, we performed similar time-course experiments following the injection of control or clodronate liposomes to evaluate the timing needed to initiate phagocyte depletion. When granulocyte numbers were assessed by proxy via qPCR (Supplementary Fig. 6), there was no effect at 6 h post-injection, yet by 8 hours there was a significant and sustained reduction in *eater* and *Nimrod B2* transcripts, indicative of granulocyte depletion (Supplementary Fig. 6). Together, these data suggest that liposome uptake occurs within 8 hours post-injection and that liposomes are quickly processed to promote phagocyte depletion. Since previous studies only evaluated phagocyte depletion at 24 or 48 h post-injection[28,29,32,33,44], these data provide greater resolution into the timing of liposome processing, enabling a more precise evaluation of candidate genes identified in our clodronate CRISPR screen.

## In vivo validation of candidate genes

From the results of the clodronate CRISPR cell screen (Fig. 2), we next identified candidates for further validation in vivo in *An. gambiae* hemocytes. To do this, the top 50 hits from each replicate (of which 12 genes were identified in both screens) were cross-referenced with scRNA-seq of *An. gambiae* hemocytes[44] to confirm their expression in mosquito granulocyte populations (Fig. 3a). Based on their presence in both screens and predicted function (Fig. 2c, Supplementary Data 6), a total of 10 candidates were selected for further validation in vivo (Fig. 3a) using RNA interference (RNAi). We performed dsRNA injections for all 10 candidate genes, resulting in successful knockdown of 5 of the 10 genes (*Tsp3A*; *PGAP6*, AGAP002672; *Traf6*, AGAP003004; *GstD3*, AGAP004382; *TMEM147*) at two days post-injection (Fig. 3b). Additional experiments to examine gene-silencing at four days post-injection for the remaining candidates similarly failed to induce a knockdown (Supplementary Fig. 7). We did not observe distinguishable differences in gene expression or hemocyte enrichment between successful and non-successful candidates for in vivo RNAi experiments (Supplementary Fig. 8).

To confirm candidate gene function in clodronate liposome-mediated phagocyte depletion, RNAi was performed in adult female mosquitoes before injection with control or clodronate liposomes. The influence of gene-silencing on clodronate-mediated granulocyte depletion was then evaluated at 8 or 24 h via the expression of *eater* and *Nimrod B2* (Fig. 3c). Clodronate liposome treatment significantly reduced *eater* at 8 hours post-injection and both *eater* and *Nimrod B2* expression at 24-hours post-injection in the control dsGFP background (Fig. 3d), while silencing of *Tsp3A*, *PGAP6*, *Traf6*, *GstD3*, and *TMEM147* each impaired phagocyte depletion, resulting in higher expression levels of *eater* and *Nimrod B2* compared to controls (Fig. 3d). We observed differences in the ability of RNAi to impair phagocyte depletion amongst the five candidate genes examined, with only *PGAP6* and *GstD3* able to completely inhibit the effects of clodronate liposome treatment at 8- and 24-hours for both reporter genes (Fig. 3d). Together, these phenotypes confirm the role of each candidate gene in clodronate liposome-mediated phagocyte depletion.

## Liposome uptake is mediated by phagocytosis

To better understand the roles of our candidate genes and the uptake mechanisms of clodronate liposomes in invertebrate cells, we first examined the influence of endocytic pathways on liposome uptake. To assess the role of either endocytosis or phagocytosis on liposome uptake, mosquitoes were intrathoracically injected with pharmacological inhibitors targeting endocytosis (chlorpromazine, CPZ)[50–52] or phagocytosis (cytochalasin, CytoD)[53–56] (Fig. 4a), with the injection of 10% DMSO serving as a control. When mosquitoes were challenged with LP-DiO particles following inhibitor treatment, the uptake of LP-DiO particles was only significantly impaired in mosquitoes treated with CytoD (Fig. 4b), suggesting that liposome uptake is dependent on immune cell phagocytosis. Additional experiments confirm that CytoD treatment impairs phagocyte depletion (Fig. 4c), demonstrating that phagocytic function is integral to clodronate liposome-mediated phagocyte depletion.

After demonstrating that CytoD treatment impedes phagocytosis in vivo (Fig. 4d), we sought to address whether any candidate genes identified in the CRISPR screen may similarly influence phagocytosis and liposome uptake. When phagocytosis experiments were performed following RNAi-mediated gene silencing, only the *Traf6*-silenced background displayed notable defects in phagocytic ability (Fig. 4e). This suggests that the impairment of clodronate liposome-mediated phagocyte depletion by *Traf6* RNAi (Fig. 3a, d, Supplementary Data 6) is likely mediated through phagocytic function (Fig. 4f). Moreover, the minimal influence of the remaining candidate genes on phagocytosis suggests that their function lies downstream of liposome uptake.

## Candidate genes that impair clodronate liposome processing are involved in phagolysosome formation

Generally, phagocytosis of extracellular particles results in formation of an early phagosome which undergoes maturation and ultimately fuses with the lysosome to form a phagolysosome, facilitating pathogen killing and protein degradation[57–59] (Fig. 5a). To better understand how clodronate liposomes are processed following phagocytosis and to identify potential roles of our candidate genes in this process, we utilized LP-DiO particles to visualize liposome uptake and processing in mosquito hemocytes. Approximately 8 h post-injection, LP-DiO particles colocalize with lysosomes (Fig. 5b), indicating that the normal processing of liposome particles involves the formation of the phagolysosome (Fig. 5a). In addition, we observed distinct patterns of DiO localization in immune cells, with some cells displaying punctate DiO localization, suggesting the presence of intact LP-DiO particles (referred to as LP-DiO⁺ cells, Fig. 5c), or those that displayed a more diffuse pattern of DiO, suggesting the breakdown and release of the LP-DiO particles (referred to as DiO⁺ cells, Fig. 5d). We found that both

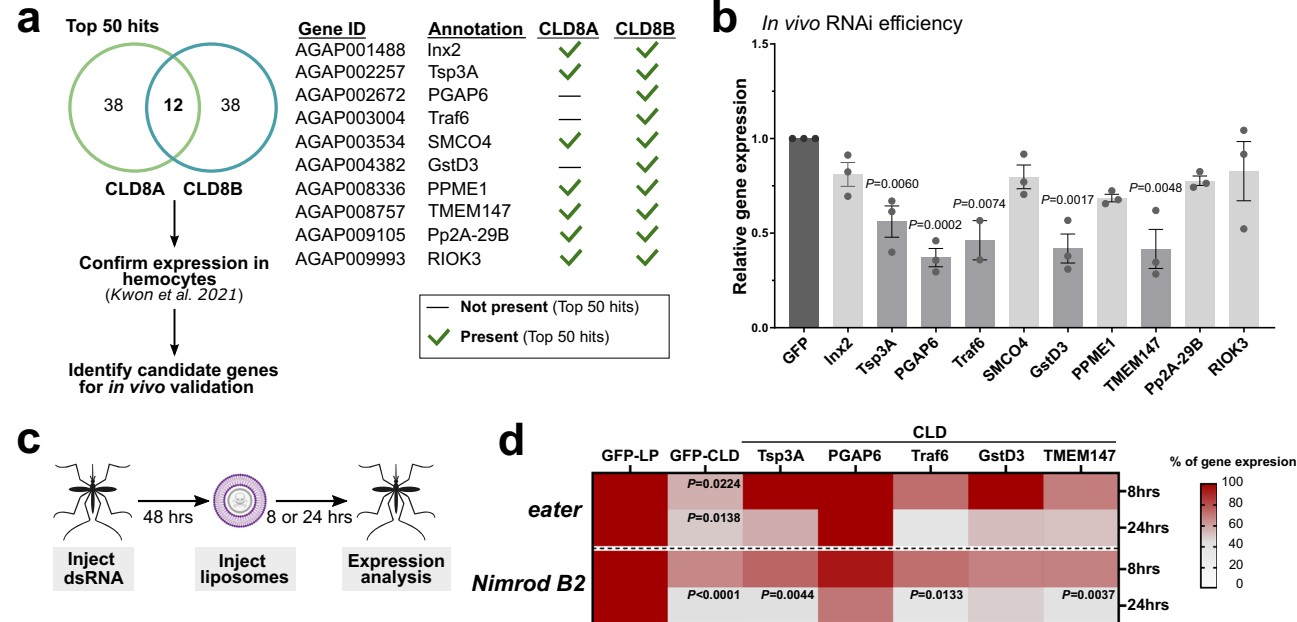

**Fig. 3 | RNAi and in vivo validation of candidate genes involved in clodronate liposome function. a** Genes identified from both clodronate liposome screens were selected based on their enrichment in both screens, expression in mosquito hemocyte populations[44], and presumed biological function to select for candidate genes for further validation in vivo. Following the injection of gene-specific dsRNAs, the efficiency of RNAi was evaluated in whole mosquitoes via qRT-PCR two days post-injection (**b**). Expression data from three independent experiments are displayed as the mean ± SEM and compared to GFP controls. Statistical differences were examined using a two-tailed unpaired t-test for each individual gene and compared to controls. Bar graph colors denote differences between the control (dark grey), statistically significant (medium grey), and non-significant (light grey) genes. Exact *P* values are displayed in the figure where significant. Additional details of the statistical analysis are included in the Source Data file. **c** To determine the influence of candidate genes on clodronate liposome function, candidate genes were first silenced via the injection of dsRNAs, then control or clodronate liposomes were injected two days post-dsRNA injection. The effects of gene-silencing

on clodronate liposome function were assessed by the expression of *eater* and *Nimrod B2* as a proxy to measure immune cell depletion (**d**). The heatmap summarizes the effects of gene-silencing on the efficacy of clodronate liposome-mediated cell ablation at 8 and 24 h, where non-significant changes in *eater* and *Nimrod B2* expression support that the gene-silenced background impairs clodronate liposome function. Data represent three or more independent experiments. For each RNAi background, expression data were compared between control liposomes and clodronate liposomes. Differences in relative gene expression were examined between GFP controls and each RNAi background using mixed-effects analysis and a Dunnett's multiple comparison test. Adjusted *P* values are displayed in the figure where significant. Additional details of the statistical analysis are included in the Source Data file. Images in **a** and **c** were created in part using BioRender. Smith, R. (2025) https://BioRender.com/l758x5w and incorporate illustrations created by David Hall using Inkscape. Source data are provided as a Source Data file.

*Tsp3A* and *Traf6* RNAi displayed a significant increase in the accumulation of LP-DiO+ cells (Fig. 5c). Conversely, silencing of *Tsp3A*, *PGAP6*, and *TMEM147* significantly reduced the percentage of DiO+ cells (Fig. 5d), suggesting that these RNAi backgrounds were impaired in their ability to breakdown LP-DiO+ particles. Together, these data suggest that our candidate genes, with the exception of *GstD3*, contribute to the internal processing of liposome particles likely through the formation of the phagolysosome.

To further validate this phenotype, we performed additional experiments using Bafilomycin A1 (BAF A1), an inhibitor of lysosome acidification and phagolysosome formation (Fig. 5a). Similar to the observed DiO localization phenotypes (Fig. 5c and d), BAF A1 treatment significantly increased the percentage of LP-DiO+ cells, while reducing the percentage of DiO+ cells (Fig. 5e). Additional experiments to evaluate clodronate liposome function in the BAF A1-treated background demonstrated that BAF A1 significantly inhibits clodronate liposome-mediated phagocyte depletion (Fig. 5f). These phenotypes are strikingly similar to the *Tsp3A*-silenced background, as well as the partial phenotypes associated with *PGAP6, Traf6*, and *TMEM147* RNAi, supporting the hypothesis that the above candidate genes are required for phagolysosome formation (Fig. 5g).

Together, these data support a model in which the phagocytic uptake of liposomes involves Traf6 and can be inhibited by CytoD treatment (Fig. 6). Additionally, the knockdown of several genes, such as *Tsp3a*, *PGAP6*, and *TMEM147*, mimics the effect of the BAF A1

inhibitor, indicating their role in further liposome processing and phagolysosome formation (Fig. 6). Although silencing of *GstD3* influenced clodronate liposome function (Fig. 3), experiments examining liposome uptake and processing did not yield phenotypes for *GstD3*, suggesting that GstD3 contributes to the downstream events that promote cell ablation (Fig. 6).

## Discussion

Forward-genetic CRISPR knockout screens enable an unbiased interrogation of gene function across a wide range of biological topics[60]. Although evidence has demonstrated the utility of this forward-genetics approach from mammals[61,62] to *Drosophila*[16,19], this methodology has not yet been extended to insect systems of public health relevance. Building on our earlier proof-of-principle CRISPR knockout screen in mosquitoes[21], we herein perform the first genome-wide pooled CRISPR screens in *Anopheles* to identify genes involved in host fitness and to uncover mechanisms of clodronate liposome function.

Our genome-wide CRISPR knockout fitness screen identified putative fitness genes that are likely required for *Anopheles* Sua-5B cell growth, division, and/or viability (Supplemental Fig. 1 and Supplemental Data 2). The *Drosophila* orthologs of most of these also scored as fitness genes in *Drosophila* cells[18] and encode proteins involved in fundamental cell functions. Furthermore, some of these overlap with essential genes identified in human cells[47]. These findings confirm the quality of the screening platform developed for *Anopheles* and help to

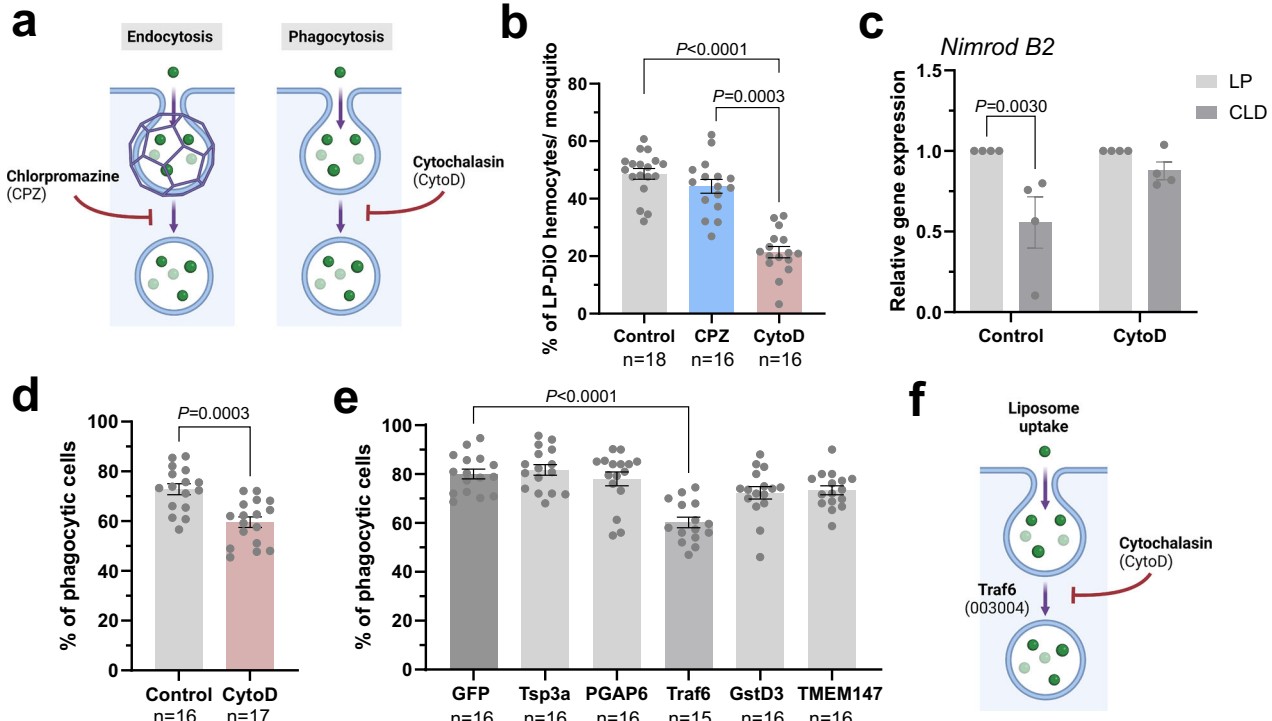

**Fig. 4 | Clodronate liposome uptake is mediated by phagocytosis. a** Overview of endocytic pathways, clathrin-mediated endocytosis and phagocytosis, with their respective inhibitors. To address the manner by which liposomes undergo uptake, mosquitoes were injected with the respective endocytic inhibitors (**a**), with the uptake of LP-DiO particles by mosquito hemocytes (immune cells) assessed at 8 h post-injection as the percentage of LP-DiO⁺ hemocytes (**b**). Data were collected from individual mosquitoes (n; dots) from two independent replicates and examined using Kruskal-Wallis with a Dunn's multiple comparisons test. Data are displayed as the mean ± SEM with exact *P* values are denoted in the figure where significant. Additional details of the statistical analysis are included in the Source Data file. **c** The effects of phagocytosis-inhibition (via cytochalasin D, cytoD) on clodronate liposome efficacy were evaluated using *Nimrod B2* expression as a proxy for immune cell ablation. Data from four independent experiments are displayed as the mean ± SEM and were analyzed using multiple unpaired t-tests with a two-stage step-up correction to determine significance. Exact *P* values are displayed in the figure where significant. Additional details of the statistical analysis are included in

the Source Data file. After further confirmation that cytoD treatment reduces the uptake of fluorospheres (**d**), candidate genes were evaluated for potential phenotypes that similarly influence phagocytosis (**e**). Data from **d** and **e** display the mean ± SEM values collected from individual mosquitoes (n) from two independent replicates. Statistical analysis was respectively performed with individual (two-sided Mann-Whitney) or multiple comparisons (Kruskal-Wallis and Dunn's multiple comparisons test) to determine significance. Exact *P* values are displayed in the figure where significant. Additional details of the statistical analysis are included in the Source Data file. Bar graph colors in **e** denote differences between the control (dark grey), statistically significant (medium grey), and non-significant (light grey) genes. These data contribute to a model (**f**) suggesting that liposome uptake is mediated by phagocytosis and implicates Traf6 (with AGAP gene ID number in parenthesis) in phagocytic uptake. Summary figures in **a** and **f** were created using BioRender: Smith, R. (2025) https://BioRender.com/hfec788; Smith, R. (2025) https://BioRender.com/ljy5a6i. Source data are provided as a Source Data file.

define a set of fitness-related genes in mosquitoes that are partially conserved across metazoa.

A subset of *Anopheles* fitness genes (146) have *Drosophila* orthologs that were not previously detected as fitness genes in *Drosophila* S2R+ cells (Supplementary Data 3). This discrepancy may reflect broader interspecies differences, or more simply, gaps in ortholog mapping. Despite these limitations, we were interested to explore this gene list. The list includes genes involved in chromatin regulation and transcription, as well as several components of the PIWI pathway (argonaute 3, AGAP008862; piwi, AGAP009509; and aubergine, AGAP011204). Although *Anopheles* and *Drosophila* share a conserved core set of PIWI-family genes, this gene family has undergone substantial expansion in mosquitoes[63], suggesting that its roles may be more prominent and functionally diverse in mosquito lineages. Moreover, unlike in *Drosophila*, where PIWI activity is largely restricted to the germline, mosquitoes exhibit somatic expression of PIWI genes, where the piRNA pathway is active in a range of somatic tissues and cell types[64]. Although less well characterized in *Anopheles*, somatic piwi expression may support essential processes such as transposon silencing, maintenance of genome integrity, and possibly antiviral immunity. Therefore, a fitness requirement for PIWI genes in mosquito cell lines may reflect a broader, species-specific function that is either

absent or dispensable in *Drosophila* S2R+ cells. It will be interesting in the future to more fully explore potential mosquito-specific gene dependencies, particularly as these might be driven by their ecological niches, blood-feeding behaviors, and roles as disease vectors.

Among fitness-related genes that are in common between *Anopheles* and *Drosophila*, we identified an ortholog of *serpent* (*srp*), which encodes a GATA transcription factor critical for hematopoiesis in *Drosophila*, and found that in vivo knockdown in *An. gambiae* reduced hemocyte numbers and increased malaria parasite infection, suggesting a conserved role in immunity. Additionally, we identified zfh1 (AGAP000779) as a transcriptional regulator necessary for Sua-5B cell fitness that was enriched in hemocytes at the bulk transcriptomic level, although not unique to a specific hemocyte subtype based on single-cell data. Given that its human ortholog ZEB1 and the *Drosophila* homolog are involved in hematopoietic differentiation, zfh1 may represent a lineage-specific regulator of mosquito hemocyte function that is deserving of further investigation. Together, these data highlight the potential of our newly generated fitness gene dataset to provide new insight into mosquito immune biology and hematopoiesis.

The fitness gene dataset identified in this work for *Anopheles* may also lead to the development of new approaches for mosquito control,

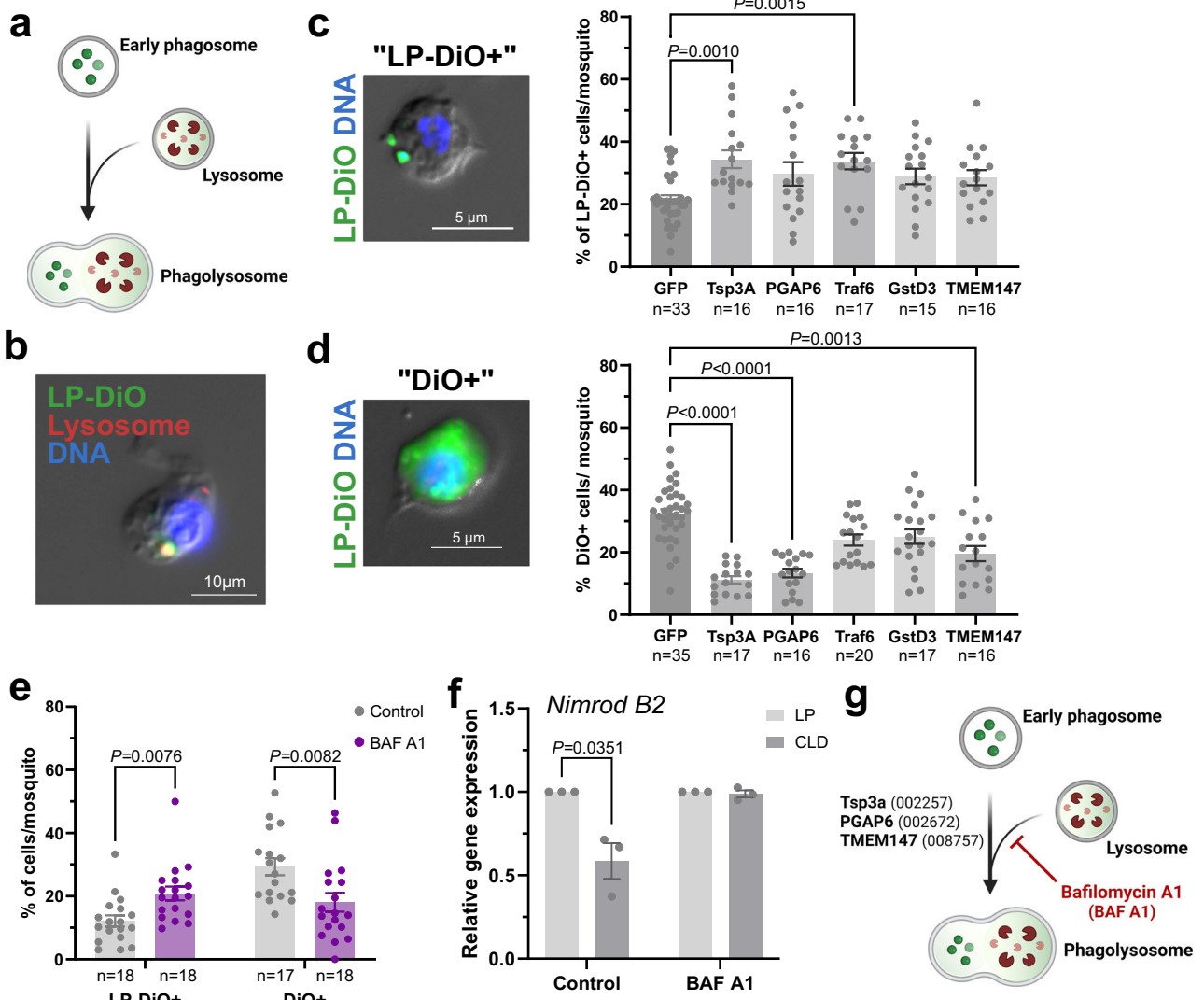

**Fig. 5 | Clodronate liposome processing requires candidate genes involved in phagolysosome formation. a** Overview of phagosome maturation and phagolysosome formation after fusion with the lysosome. To confirm that phagocytosed liposome particles undergo phagolysosome formation, immunofluorescence assays were performed following LP-DiO injection and staining with lysosome-specific dye, LysoView 594 (**b**). Co-localization of LP-DiO particles (green) and the lysosome (red) support that liposomes are processed by the formation of the phagolysosome prior to degradation. Observations of LP-DiO⁺ cells display two distinct phenotypes, where LP-DiO particles are punctate and remain intact (referred to as LP-DiO⁺; **c**), or where DiO fluorescence is diffused (referred to as DiO⁺) suggesting that liposome particles have been degraded (**d**). These LP-DiO⁺ (**c**) and DiO⁺ (**d**) phenotypes were evaluated in the gene-silenced backgrounds for each of the candidate genes identified in the CRISPR screen. Data were collected from individual mosquitoes (n; dots) from two or more independent replicates. For **c** and **d**, data are displayed as the mean ± SEM and examined using Kruskal-Wallis with a Dunn's multiple comparisons test. Exact *P* values are displayed in the figure where significant. Additional details of the statistical analysis are included in the Source Data file. For both **c** and **d**, bar graph colors denote differences between the control (dark grey), statistically significant (medium grey), and non-significant (light grey) genes. **e** To further refine these observed phenotypes, we evaluated LP-DiO⁺ and DiO⁺ phenotypes following treatment with Bafilomycin A1 (BAF A1), an inhibitor of lysosome fusion with the phagosome. Data are displayed from individual mosquitoes (n; dots) from two independent experiments. **f** The effects of BAF A1 inhibition on clodronate liposome function were evaluated from three independent experiments using *Nimrod B2* expression as a proxy for immune cell ablation. Data in **e** and **f** are displayed as the mean ± SEM and analyzed using multiple unpaired t tests to determine significance and corrected using Holm-Šídák for multiple comparisons. Adjusted *P* values are displayed in the figure where significant. Additional details of the statistical analysis are included in the Source Data file. These data contribute to a model (**g**) suggesting that liposome processing is mediated by formation of the phagolysosome involving Tsp3A, PGAP6, and TMEM147, which can be impaired using the inhibitor BAF A1. Summary figures in **a** and **g** were created using BioRender: Smith, R. (2025) https://BioRender.com/43qzq8b and Smith, R. (2025) https://BioRender.com/2gttspd. Source data are provided as a Source Data file.

where fitness genes may serve as targets for population suppression strategies that aim to reduce or eliminate mosquito populations[65]. For example, one or more genes determined to be essential through future in vivo studies in *Anopheles*, could be genetically targeted to create a synthetic gene-drive system capable of promoting mosquito lethality. This has the potential to enhance population replacement strategies relying on CRISPR[66,67], homing endonuclease[68], Medea-like[69,70], or

cleave and rescue[71,72] technologies as a means for selection against non-replacement alleles in split-drive systems.

Clodronate liposomes have been widely used in studies of vertebrate immunology[30,31], and more recently in arthropod systems[28,32,33], to promote the targeted ablation of phagocytic immune cell populations. While evidence suggests that clodronate-derived metabolites act as ATP analogs to block mitochondrial ATP synthase activity and

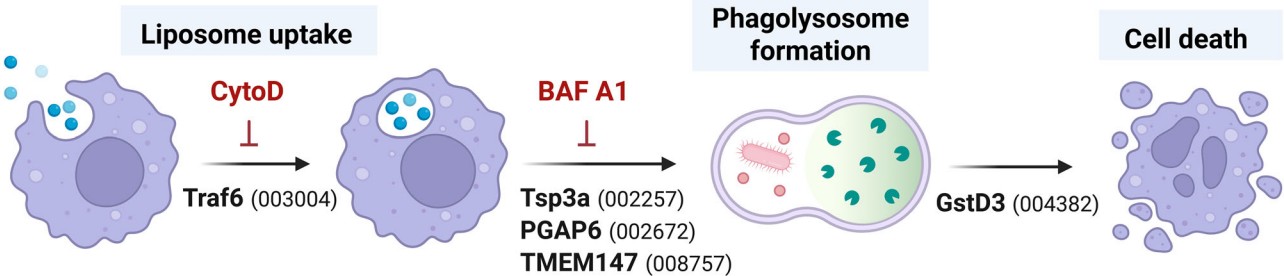

**Fig. 6 | Summary of candidate genes involved in clodronate liposome function.** Experiments support a model in which candidate genes identified in the CRISPR screen contribute to the uptake, processing, and downstream function of clodronate liposomes in promoting immune cell ablation in *An. gambiae*. Created in BioRender. Smith, R. (2025) https://BioRender.com/16sab2o.

consequently trigger apoptosis[73], the precise mechanisms of clodronate liposome uptake and processing have not been adequately explored. The results of our genome-wide screen provide a comprehensive examination of clodronate liposome function in *Anopheles*, implicating a set of 88 genes identified in one or both of our screens. This gene set is enriched for genes involved in cellular metabolism and methyltransferase function, bringing new mechanistic insights into clodronate liposome function.

Additionally, we can infer subcellular components and pathways involved in clodronate liposome function based on functions of orthologs of genes identified in the screen. For example, multiple hits are components or regulators of the serine/threonine-protein phosphatase 2 A (PP2A) protein complex, which is involved in a variety of biological processes such as cell growth, differentiation, apoptosis, and immune regulation[74]. Moreover, *PPME1* is a methyl-esterase enzyme that acts directly on the catalytic subunit by demethylation of the PP2A protein complex to cause its inactivation[75]. Two other hits contribute to the regulation of the same protein complex: *LCMT1* (AGAP008768), a methyl-transferase enzyme responsible for the methylation of PP2A at the same site targeted by *PPME1*[76], and *MASTL* (AGAP001636), which acts indirectly to promote the inactivation of PP2A[77]. Of note, both *Pp2A-29B* and *MASTL* are fitness genes in both human (DepMap) and *Anopheles* cells, and are among the most highly enriched targets in both our clodronate screens. This suggests that, although the knockout of these genes impacts cell fitness under "normal" conditions, their knockout provides a growth advantage under clodronate selection, as the cells become less sensitive to the drug compared to normal cells. In addition, we found multiple hits in the clodronate liposome screen that correspond to serine/threonine-protein kinase signaling. These include *RIOK3*, a serine/threonine-protein kinase[78], and *RPS17*, a component of the 40S ribosomal subunit that directly interacts with *RIOK3* during ribosome biogenesis[79]. This suggests that these genes may be involved in the same signaling pathway and potentially implicates ribosome biogenesis in clodronate liposome function. Further experiments are needed to establish how PP2A and *RIOK3* signaling components are related to clodronate liposome function.

Through the use of pharmacological inhibitors in vivo, we demonstrate that the cellular uptake of clodronate liposomes by mosquito hemocytes is mediated by phagocytosis, and not by clathrin-mediated endocytosis, providing further support for the specificity of clodronate liposomes to explicitly target phagocytic immune cells[80]. In addition, knockdown of one candidate identified in our clodronate screen, *Traf6*, resulted in notable defects in phagocytosis, supporting that *Traf6* likely influences uptake of clodronate liposomes. However, since Traf6 is a RING-type ubiquitin ligase that interacts with several immune signaling molecules[81,82], these effects are likely indirect, and possibly resulting from impaired production of downstream immune effectors or defects in immune cell activation[83].

Additional microscopy, RNAi, and inhibitor experiments performed in vivo confirm that the formation of the phagolysosome is a critical step in the processing of clodronate liposomes. Lysosomes contain various hydrolytic enzymes that promote the breakdown of macromolecules for degradation and cellular recycling[84], thereby serving an essential role in the breakdown of the liposome particle and the intracellular delivery of clodronate required to initiate cell death. Taking advantage of the fluorescence of LP-DiO particles, we demonstrate the co-localization of liposome particles with the lysosome, as well as the punctate and diffuse patterns of DiO that enable the visualization of liposome processing. We demonstrate that three candidate genes identified in our CRISPR screen, *Tsp3A*, *PGAP6*, and *TMEM147*, have key roles in phagolysosome formation and validate these phenotypes in liposome degradation through the use of the BAF A1 inhibitor to impair lysosome fusion. The products of these genes are believed to localize to cell membranes and have been implicated in immune cell function in orthologous systems[85–87]. The human ortholog of *PGAP6* is a GPI-anchored phospholipase predicted to localize to the lysosome[86,88], suggesting that *PGAP6* might play a role in the breakdown of liposome particles and the subsequent release of clodronate following phagolysosome formation. Other genes identified in the screen similarly suggest that lysosome fusion is an integral step in clodronate liposome processing. For example, *CLVS1* (AGAP005388) is required for proper formation of late endosomes and lysosomes[89], and although AGAP011017 is of unknown function, it harbors a putative lipid binding domain (InterPro) similar to the Ganglioside GM2 activator (GM2-AP), which acts as a lysosomal lipid transfer protein. Our results also suggest that GstD3 acts downstream of liposome intracellular processing. As a member of a large family of glutathione S-transferases involved in cellular detoxification and insecticide resistance, GstD3 may have roles in clodronate metabolism that ultimately contribute to its ability to promote apoptosis and cell ablation.

A key step of clodronate toxicity is its incorporation into AMP molecules to form a non-hydrolysable analog of ATP, the adenosine 5'β-γ-dichloromethylene triphosphate (AppCCl2p), which inhibits the mitochondrial translocase and putatively induces apoptosis through mitochondrial depolarization. It has been proposed that aminoacyl-tRNA synthetases might be responsible for the incorporation of clodronate (a bisphosphonate analog of pyrophosphate, PPi) into AMP molecules by a reverse reaction[73]. However, although the reaction in which PPi would be incorporated into ADP to regenerate ATP is theoretically possible, it is not favored due to thermodynamic and kinetic constraints. In fact, the energy released from ATP hydrolysis and the rapid degradation of PPi by pyrophosphatases ensure that the reverse process does not occur naturally[90]. As a result, aminoacyl-tRNA synthetases (aaRS) typically catalyze the forward reaction of ATP hydrolysis to charge a tRNA with an amino acid, producing AMP and pyrophosphate (PPi)[91]. The presence of a non-hydrolysable form of PPi, such as clodronate, might affect the stoichiometry of the reaction, and

one or multiple enzymes that have PPi and AMP as byproducts might perform a reverse reaction that incorporates clodronate into AMP molecules. We did not identify aminoacyl-tRNA synthetases in the screen but note that they could have been missed due to their essential roles in cell metabolism. Alternatively, there could be multiple aminoacyl-tRNA synthetases responsible for catalyzing this reaction, masking the phenotype from genetic enrichment.

In summary, the results of our genome-wide CRISPR knockout screens offer new insights into our understanding of mosquito fitness-related genes, immune cell biology, and clodronate liposome function. Based on comparative data in *Drosophila* and human cell lines, we identify a putative set of core fitness-related genes that are conserved across species that can inform further studies of mosquito fitness and viability. In addition, genes identified in the clodronate liposome screen uncover key components involved in liposome uptake, as well as intracellular trafficking and processing, which may provide further insights into general mechanisms of cellular processes relevant to host defense, autophagy, apoptosis, and immune cell function. Altogether, the genome-wide *Anopheles* screen platform we present and subsequent results from our initial CRISPR screens provide a foundation for forward-genetic studies in mosquitoes.

## Methods

### Ethics statement
All protocols and experimental procedures regarding vertebrate animal use were approved by the Institutional Animal Care and Use Committee at Iowa State University (IACUC-21–143). Mosquito rearing, handling, and infection experiments were approved by the Institutional Biosafety Committee at Iowa State University (IBC-21-062 and IBC-21-063).

### Data collection and analysis
Data collection and analysis were performed in an unblinded manner as described for each of the following experimental methodologies, a methodological constraint due to requirements for specialized expertise and personnel limitations. Although rigor was maintained through multiple independent biological replicates and objective measurements, we acknowledge that the lack of blinding has the potential to influence subjective measures and experimental outcomes.

### Cell culturing
The *Anopheles coluzzii* "screen-ready" (attP+ Cas9 + ) cell line Sua-5B-IE8-Act::Cas9-2A-Neo[1] (CVCL_B3N3, Drosophila Genomics Resource Center, stock # 334) as previously described[21]. The cell line was cultured at 25 °C in Schneider's medium (Gibco, #21720-024), 1x Penicillin-Streptomycin (Gibco, #15-140-148) and 10% heat inactivated fetal bovine serum (Gibco, #A5256801) and 500 µg/ml of geneticin (G-418 sulfate, GoldBio, #G-418-1).

### Genome-wide library design, and cloning
sgRNAs targeting the whole genome of *Anopheles gambiae* (AgamP4.12) were selected using CRISPR GuideXpress and following the previously described pipeline[21]. Briefly, all computed sgRNAs were retrieved, and the top seven sgRNAs per gene were selected based on the following criteria: minimal OTE (off-target effect) score; maximum ML (machine learning efficiency) score; and filtered to remove sgRNAs that match regions with SNPs in the *Anopheles coluzzii* Sua-5B cell line genome sequence. In addition, sgRNA designs with the *BbsI* site sequence were removed because *BbsI* is used for ligation-based cloning into the library vector. The library includes 89,724 unique gene-targeting sgRNAs as well as control and other sgRNAs, as detailed in Supplementary Table 1. The sgRNA sequences were cloned into *BbsI*-digested pLib6.4B-Agam_695 (Accession # OL312683; Addgene # 176668) using the CloneEZ service (Genscript). Cloned vector was

subsequently reamplified with a theoretical coverage of <100 times in E. cloni 10GF' ELITE Electrocompetent Cells (Lucigen) and grown in 500 mL of LB-Ampicillin media at 30 °C overnight and bacterial pellets were frozen at -80 °C. Before transfection, plasmid DNA was prepared from 50 mL pellets by midiprep (Zymo). Sequencing of the cloned plasmid library confirmed the successful cloning of >98,3% (88159/89711) of designed sgRNAs, detectable with at least one read/sgRNA (circa 94% of guides are detected with at least 10 reads/guide and about 1.7% were lost stochastically). The library obtained was named "GWCRISPKO_AGAM" and deposited at Addgene (Addgene # 234477).

### Negative selection CRISPR screen to identify fitness genes
Sua-5B-IE8-Act::Cas9-2A-Neo cells in the log phase of growth were seeded at 35 x 10⁶ cells per 100 mm dish in growth media containing antibiotics. They were transfected with a plasmid mixture containing equimolar amounts of HSP70-ΦC31-Integrase plasmid (pBS130) and sgRNA donor plasmid library (pLib6.4B-Agam_695) using Effectene (Qiagen) according to the manufacturer's base protocol ("1:25"). We achieved a coverage of ~ 244 cells/sgRNA by transfecting 735 x 10⁶ cells [90208 sgRNAs x 244 cells/sgRNA x 0.03 (RMCE efficiency) = 735 x 10⁶] in 21,100 mm dishes. After 4 days, each dish was expanded into 2 x 150 mm dishes containing 5 mg/mL puromycin. Cells were cultured for an additional 26 days with media changes and re-seeding every 4 days. Re-seeding at each passage was maintained at a density above 1000 cells/sgRNA to ensure representation of KO pool diversity. Cells were cultured up to 60 days (8 weeks) after transfection. Following selection, genomic DNA was extracted from cell pellets containing >1000 cells/sgRNA using the Quick-gDNA MaxiPrep kit (Zymo). Next, the genomic DNA was barcoded and Illumina sequencing adapters were added via 2-step PCR amplification. Amplicon sequencing was performed using a NextSeq500 at the Biopolymers Facility at Harvard Medical School (RRID:SCR_007175). Demultiplexing and trimming of barcode labeling was performed using in-house scripts. sgRNAs with a low-read count (<10 reads in the plasmid library) were removed from the readcount files. For identification of base fitness genes, the plasmid library vector readcounts from cells after 60 days post-transfection were analyzed with MAGeCK MLE (version 0.5.6) to infer MLE Z-scores for each gene. The sgRNAs with low representation in the plasmid library (readcounts <135) were trimmed prior to MLE analysis. The list of genes effectively targeted in the fitness screens are available in Supplementary Data 1.

### Comparison of Sua-5B cells to hemocyte gene expression enrichment analysis
Gene expression in Sua-5B cells was determined based on bulk RNA-seq data as described above, considering genes with a log10(TPM + 0.01) ≥ 1 as expressed. The totality of *Anopheles* annotated genes was considered in the analysis as the background gene set. To assess enrichment of Sua-5B expressed genes within different hemocyte clusters and Bulk RNA-seq derived from published datasets[43,44], both empirical randomization tests and analytical hypergeometric tests were performed. For each sample/cluster (Bulk RNA-seq, HC1–HC6, FBC1–FBC2, MusC from Raddi et al.[43] and C_1–C_8 from Kwon et al.[44]), the observed number of genes expressed in Sua-5B was calculated. To ensure robustness of the analysis we used two approaches: a randomization analysis and Hypergeometric test. For the randomization analysis for each sample, random sets of genes (equal in number to the sample size) were drawn without replacement from the background genome, performing 100 K iterations (drawings) for the clusters and 1 M iterations for the Bulk RNA-seq sample. Using this analysis, we obtained an empirical *p*-value, calculated as the proportion of random samples producing an equal or greater number of expressed genes than observed. Random number generation was performed without fixing the random seed. The mean overlap from randomizations was recorded. For the hypergeometric test (a mathematical analysis), the

probability of observing the measured or greater overlap was calculated using the cumulative hypergeometric distribution, considering the size of the genome, the number of expressed genes, and the sample size. Summary tables including sample size, observed overlap, mean random overlap, empirical p-value, hypergeometric *p*-value, and number of iterations were compiled. All analyses were performed using Python v3.12, with the following packages: pandas (v2.2.2); numpy (v1.26.4); matplotlib (v3.8.4); scipy (v1.12.0). All analyses were conducted in a JupyterLab environment to ensure reproducibility and transparency in data handling. Scripts for data wrangling, randomization analysis and calculation of hypergeometric mean were generated with the assistance of AI (ChatGPT-4o).

## Ortholog mapping and comparison with fitness genes in *Drosophila*
Mapping of *Anopheles* genes to *Drosophila* and to human ortholog was done using DIOPT[36] (v 9.0). Ortholog mapping was filtered based on DIOPT rank (only high or moderate rank excluding low rank mapping) and the orthologs of the fitness genes in Anopheles were compared with the corresponding data from *Drosophila* or human respectively. Comparisons with a comparable data set from a *Drosophila* CRISPR cell screen were based on MLE Z values from a previous CRISPR screen in S2R$^+$ cells (at the same 5% FDR)[18]. Comparisons with human data were performed using a list of core essential genes identified from human cell lines[47]. The fitness genes in *Anopheles* (Supplementary Data 2) are compared with *Drosophila* fitness gene and human essential gene lists in Supplementary Data 3.

## Gene set enrichment analysis
To perform gene set enrichment analysis (GSEA), *Drosophila* orthologs mapped as described above from mosquito genes that scored as fitness genes (1280 genes) or ranked within the first fifty hits in the two clodronate liposome screens (88 genes from Clodronate A & B) were used as input for analysis with PANGEA[37]. For fitness gene orthologs, gene set enrichment analysis of the replicate 1 fitness gene list was based on generic gene ontology (GO) slim biological process (BP) terms[38]; Gene List Annotation for Drosophila (GLAD) gene groups[39], or FlyBase phenotype annotations for classical mutations[40], and the full sets of outputted enrichment data from PANGEA are included in Supplementary Data 4. For clodronate liposome screen hit analysis, the same three gene sets were used, and these were supplemented by additional analysis using the *Drosophila* GO BP and FlyBase Gene Group gene sets (Supplementary Data 6). The specific selections made at the PANGEA user interface are indicated on the first row of the PANGEA analysis sheets within Supplementary Data 4 and Supplementary Data 6.

## Positive selection CRISPR screen with clodronate liposomes
For the positive selection screen, 30 days post library transfection cells were selected in media containing puromycin and 16 μM liposome as a control or 8 μM clodronate. The concentrations used in the screen were established for Sua-5B-IE8-Act::Cas9-2A-Neo cells to be close to the IC$^{50}$ for the clodronate (IC$^{50}_{Clodronate}$ = 7.4 μM) and negligible for the liposome vehicle (IC$^{50}_{Liposome}$ = 81.6 μM), as established by assaying total ATP levels (indirect readout of cell growth) during a 6-day treatment in 96-well format using the CellTiter-Glo assay (Promega), following the manufacturer's recommendations (outlined in Fig. 2a). To validate that these concentrations produced consistent phenotypes at larger scale, we also confirmed the IC$_{50}$-derived phenotypes in 150 mm dishes ahead of the genome-wide screen. For the screen the cells were selected through three cycles of treatment. Each cycle of treatment consisted of seeding the cells in media with liposome vehicle or clodronate liposome, followed by media change and re-seeding two additional times. Except in the case of the treatment "Clodronate B," in which the cells were exposed to selective media a single time for the first 4 days and then allowed to recover with normal media before the next cycle, all the other treatments were performed by continuous exposure to the selective media. The cells were re-seeded at a density above 1000 cells/sgRNA at each passage to ensure representation of KO pool diversity. Following selection, genomic DNA extraction, barcoding, sequencing and analysis was performed as detailed above. Readcount and data analysis, including enrichment analysis and Robust Rank Aggregation score calculation, were performed using MaGeCK 0.5.7 and scatter plots were visualized with Prism (v 10.1.0). The list of genes effectively targeted in clodronate liposome screens are available in Supplementary Data 1.

## Mosquito rearing
*Anopheles gambiae* mosquitoes (Keele strain)[92] were reared at 27 °C and 80% relative humidity, with a 14:10 h light: dark photoperiod. Larvae were fed on commercialized fish flakes (Tetra), while adults were maintained on a 10% sucrose solution and fed on commercial sheep blood (Hemostat) for egg production.

## RNA isolation and gene expression analyses
RNA isolation from whole adult mosquitoes was performed using TRIzol (Invitrogen) according to the manufacturer's protocol. Two micrograms of total RNA were used for first-strand synthesis using the LunaScript RT SuperMix Kit (NEB, #E3010L). Gene expression was analyzed with quantitative real-time PCR (qPCR) using PowerUp SYBRGreen Master Mix (Thermo Fisher Scientific, #A25742), while results were analyzed using the 2$^{-\Delta Ct}$ method and normalized against the internal reference, *rpS7*, as previously described in ref. 26,28,93. All qPCR primers are listed in Supplementary Table 2.

## Timing experiments examining the uptake of fluorescent liposome particles
To determine the approximate timing of liposome uptake in vivo, mosquitoes were injected with 69 nl of Fluoroliposome-DiO (LP-DiO, Encapsula Nano Sciences, #CLD-8912) using a 1:50 dilution in 1X PBS. After injection, mosquitoes were incubated at 27 °C for 1, 2, 6, 8, or 12 h, then were injected with a suspension containing 200 μM of Vibrant CM-DiI (Thermo Fisher Scientific, #V22888) and 2 mM of Hoechst 33342 (Thermo Fisher Scientific, #62249) to label mosquito hemocytes. After an additional incubation of 30 min at 27 °C to enable in vivo staining, hemolymph was perfused from each mosquito using an anticoagulant buffer of 60% v/v Schneider's Insect medium, 10% v/v fetal bovine serum (FBS), and 30% v/v citrate buffer (98 mM NaOH, 186 mM NaCl, 1.7 mM EDTA, and 41 mM citric acid; buffer pH 4.5) as previously described in refs. 26–29,44. Hemolymph perfusions were placed directly on multi-well microscopic slides for downstream analysis by microscopy. Cells were allowed to adhere for 20 min and fixed with 4% paraformaldehyde (PFA) for 10 min, followed by five washing steps with 1X PBS. Samples were observed under a Zeiss fluorescent microscope to calculate the percentage of hemocytes (of total) taking up the fluorescent LP-DiO particles.

## Phagocyte depletion with clodronate liposomes
Naïve mosquitoes were injected with either clodronate (CLD) or control (LP) liposomes (Encapsula Nano Sciences, #CLD-8901) to deplete phagocytic immune cell populations in *Anopheles gambiae* as previously described in refs. 28,94. Based on the demonstrated IC$_{50}$ of clodronate liposomes in vitro as part of this study, liposomes were diluted to a similar concentration using a 1:50 dilution in 1X PBS for all in vivo studies herein. Previous studies were performed using a more concentrated 1:5 dilution[28,29,44]. Final concentrations of clodronate liposomes were calculated based on a hemolymph volume of ~2 μl[95]. To determine the approximate time needed for phagocyte depletion, mosquitoes were injected with either 69 nl of CLD or LP, and then incubated at 27 °C for 6, 8, 12, or 24 h. Whole mosquito samples were

then processed for RNA isolation and cDNA synthesis as described above. The expression levels of *Eater* and *Nimrod B2* were used as a proxy to demonstrate phagocyte (granulocyte) depletion[28,29,44,94].

To evaluate the effects of phagocyte depletion over time, 3-day old female *Anopheles gambiae* mosquitoes were injected with either control liposomes (LP) or clodronate liposomes (CLD) at a 1:50 dilution. Mosquitoes were maintained under controlled conditions at 27 °C with 80% relative humidity and a 14:10 h light: dark photoperiod. Every three days (1, 4, 7, and 10 days post-treatment), the proportion of phagocytes (granulocytes) was quantified using a hemocytometer as previously described[25,28,94].

## dsRNA synthesis and gene-silencing

Candidate genes identified in the genome-wide CRISPR screen were validated using RNAi-mediated gene silencing to confirm their functional roles in the mode of clodronate action. T7 primers specific to each gene (Supplementary Table 3) were used to amplify DNA templates from whole female mosquito cDNA samples to synthesize long dsRNAs using the MEGAscript RNAi kit (Thermo Fisher Scientific, #AM1626). Following synthesis, the concentration of dsRNAs was adjusted to 3 µg/µl. Adult female mosquitoes (3–5 days old) were cold anesthetized and injected with 69 nl of dsRNA targeting each candidate gene. For each experiment, mosquitoes were also injected with dsRNA targeting GFP as a negative control. All injections were performed using Nanoject III (Drummond Scientific). Gene-silencing efficiency was evaluated by qPCR at either 2- or 4-days post-injection. All experiments were performed in three or more independent biological replicates.

Efforts to examine the efficacy of RNAi in candidate genes was performed using existing data to examine the enrichment of gene expression in hemocytes and non-hemocyte tissues[43,96]. Additional analysis was performed using data from Kwon et al.[44] to determine potential correlations between the efficacy of candidate gene RNAi (this study) and averaged gene expression across all hemocyte subtypes (C2-C8) or granulocyte populations (C2-C4, C6). Relationships were determined using a Pearson correlation with GraphPad Prism software.

## Hemolymph perfusion and hemocyte counting

Hemolymph was perfused in adult female *An. gambiae* through the intrathoracic injection of an anticoagulant solution and collection of the perfusate through a small incision in the abdomen as previously described in refs. 26,27. To determine total hemocyte numbers, the collected perfusion from an individual mosquito was added to a disposable Neubauer Improved hemocytometer slide (iNCYTO C-Chip, #DHC-N01) as previously described in refs. 25,26,94.

## Malaria parasite infection

Infections with the rodent malaria model, *Plasmodium berghei*, were performed by first infecting Swiss Webster mice (Charles River) with *P. berghei*-mCherry[97] parasites as previously described in refs. 26,93. Mosquito infections were performed by allowing mosquitoes to feed on anesthetized *P. berghei*-infected mouse. Following feeding, fully engorged mosquitoes were selected by cold-sorting, then were placed at 19 °C. Oocyst numbers were evaluated by fluorescence microscopy in dissected midguts at 10 days post-infection.

## Use of inhibitors to examine liposome uptake

To examine the mechanisms of liposome uptake by mosquito hemocytes, mosquitoes were treated with 200 µM Cytochalasin D (CytoD, Sigma) to inhibit phagocytosis[53–56] or 25 µg/ml Chlorpromazine hydrochloride (CPZ, MP Biomedical, #02190326-CF) to impair clathrin-mediated endocytosis[50–52]. Mosquitoes injected with 10% DMSO in 1X PBS were used as negative controls. At 6 h post-injection, mosquitoes were injected with a 1:50 dilution of Fluoroliposome-DiO (LP-DiO) in 1X PBS, and then incubated for 8 h at 27 °C. Following injection with 2 mM Hoechst 33342 to counterstain nuclei, hemolymph

was perfused from individual mosquito samples and then observed using a fluorescent microscope to determine the proportions of hemocytes containing fluorescent liposome particles.

Additional experiments were performed to confirm the effects of CytoD on clodronate liposome uptake. Mosquitoes were first injected with 200 µM CytoD or 10% DMSO in 1X PBS and allowed to recover for 6 h at 27 °C, then followed by injection with control or clodronate liposomes (diluted at 1:50) and incubated at 27 °C for 8 h. The influence of CytoD on clodronate liposome function and resulting phagocyte depletion was evaluated by proxy through the analysis of *Nimrod B2* expression via qPCR[28,29,44].

## Phagocytosis assays

The effects of candidate genes or inhibitors on phagocytosis were evaluated by injecting adult female mosquitoes with 69 nl of 2% of green fluorescent FluoSpheres (1 µm; Thermo Fisher Scientific, #F8823) similar to previous studies[26,28]. In addition to the beads, mosquitoes were concurrently injected with 100 µM Vibrant CM-DiI and 2 mM of Hoechst 33342 in 1X PBS to counterstain hemocytes, then allowed to recover for 30 min at 27 °C. The effects of gene-silencing on phagocytosis were examined approximately 48 h after injection with dsRNAs, while the effects of the inhibitor Cytochalasin D were analyzed at 6 h post-injection to serve as a positive control to impair phagocytosis[55]. For each experiment, hemolymph was perfused from individual mosquitoes using an anticoagulant buffer and placed on multi-well microscope slides. Hemocytes were allowed to adhere for 20 min and fixed with 4% PFA. Following five washing steps, samples were mounted with Aqua Poly/Mount (Polysciences, #18606) and observed under a fluorescent microscope to determine the percentage of phagocytic cells. Approximately 50 hemocytes were counted per individual mosquito, with data collected from two or more replicates ($n = 16^+$ mosquito samples).

## Use of gene-silencing to examine liposome uptake and processing

To determine the roles of candidate genes on liposome uptake and processing, candidate genes were first silenced by the injection of dsRNA in naive adult female mosquitoes. Two days post-injection, gene-silenced mosquitoes were injected with a 1:50 dilution of Fluoroliposome-DiO in 1X PBS. Following incubation for 8 h, phenotypes were evaluated in individual mosquitoes as the percentage of hemocytes containing liposome particles (LP-DiO$^+$) to evaluate liposome uptake, or as diffused patterns of DiO (DiO$^+$) in the cytosol that support liposome processing and degradation.

## Immunofluorescence of cellular localization

To visualize the co-localization of liposome particles with the lysosome, mosquitoes were perfused with an anticoagulant buffer at 8 h post-injection with a 1:50 dilution of LP-DiO. Hemocytes were allowed to adhere for 30 min without fixation, and then incubated with the lysosome-specific dye, LysoView 594 (Biotium, #70084), using a 1:500 dilution in 1X PBS for 1 h. Samples were mounted with ProLongDiamond AntiFade Mountant with DAPI (Life Technologies, #P36971) and immediately observed using fluorescence microscopy (Zeiss Axio Imager M2).

## Use of inhibitors to impair lysosome acidification

To investigate the involvement of lysosome function in regulating the processing of clodronate-liposomes, mosquitoes were treated with 25 µM of Bafilomycin A1 (BafA1; Cayman, #88899-55-2), a proton pump V-ATPase inhibitor, or 10% DMSO in 1X PBS. Mosquitoes were incubated for 16 h at 27 °C as previously described in ref. 98, then injected with Fluoroliposome-DiO using a 1:50 dilution in 1X PBS. The number of hemocytes displaying intact liposome particles (LP-DiO$^+$) or diffused patterns of DiO (DiO$^+$) in the cytosol was determined by

immunofluorescence. To examine the effects of BafA1 on clodronate function, mosquitoes were injected with clodronate or control liposomes at 1:50 dilution in 1X PBS following treatment with Baf A1. At 8 h post-injection, phagocyte depletion was evaluated by proxy through *Nimrod B2* expression via qPCR[28,29,44].

## Software

Graphical data visualization was performed using GraphPad Prism (version 10.4.2). Microscopy images were initially captured and processed using ZEN and Zen lite software (Zeiss) prior to final processing with Adobe Photoshop (version 26.7). Figures were created using either Adobe Illustrator (version 27.8.1) or Inkscape (version 1.0.2-2) and were supplemented using images developed with BioRender (BioRender 2024).

## Reporting summary

Further information on research design is available in the Nature Portfolio Reporting Summary linked to this article.

## Data availability

All data supporting the study are included in the manuscript or its associated supplemental files. All data included in graphical outputs are available in the Source Data file. Raw data analysis associated with the CRISPR screens are provided as Supplementary Data. Sequence data used to confirm the plasmid library or for the analysis of the outcomes of our genome-wide screens are provided as unprocessed sequencing files (FASTQ) available on the Sequence Read Archive (SRA), Accession PRJNA1335561 [http://www.ncbi.nlm.nih.gov/bioproject/1335561]. Plasmids and plasmid libraries associated with the CRISPR screens are available on Addgene (#176668 and #234477). Source data are provided with this paper.

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

## Acknowledgements

We would like to thank Ian Schneider for kindly providing the cytochalasin D inhibitor. Work at the *Drosophila* Research & Screening Center-Biomedical Technology Research Resource (DRSC-BTRR) was supported by NIH NIGMS P41 GM132087 (to NP and SEM). Work at Iowa State University was supported by NIH R21 AI166857, NIH R01 AI 177540, and NIH R01 182256 (to RCS). Additional support was provided by the National Science Foundation Graduate Research Fellowship Program under Grant No. 2336877 (to D.R.H.). NP is an Investigator of the Howard Hughes Medical Institute. This article is subject to HHMI's Open Access to Publications policy. HHMI lab heads have previously granted a non-exclusive CC BY 4.0 license to the public and a sublicensable license to HHMI in their research articles. Pursuant to those licenses, the author-accepted manuscript of this article can be made freely available under a CC BY 4.0 license immediately upon publication.

## Author contributions

E.M. and G.-R.S. contributed equally to this work and are listed in alphabetical order. E.M., R.V., M.B., S.E.M., N.P., and R.C.S. designed and performed the genome-wide CRISPR screens, with E.M., R.V., Y.H. and S.E.M. contributing to bioinformatic analysis. G.-R.S., H.K., D.R.H., and R.C.S. performed in vivo experiments with *An. gambiae* and contributed to data collection and analysis. G.-R.S. performed microscopy experiments examining liposome uptake and processing. S.E.M., N.P., and R.C.S. provided supervision and experimental oversight. All authors contributed to writing and editing of the initial manuscript draft and approved the final manuscript.

## Competing interests

The authors declare no competing interests.
