## [Transparent Peer Review file · Nature Communications]

A genome-wide CRISPR screen in Anopheles mosquito cells identifies fitness and immune cell function-related genes

Corresponding Author: Dr Ryan Smith

Version 0:

Reviewer comments:

Reviewer #1

(Remarks to the Author)

This manuscript builds on a proof of principle paper published by the Perrimon group in 2022

<https://doi.org/10.1038/s41467-021-27129-3> which developed computational and biological tools and demonstrated that a genome wide CRISPR screen was possible in a mosquito hemocyte-like cell line. Here the authors apply a genome wide CRISPR screen in the *Anopheles coluzzii* hemocyte-like Sua-5b line for which they had previously generated tools to identify essential genes and those associated with immune cell function, specifically resistance to clodronate liposomes.

The authors employ a drop out approach to identify essential genes such that those sgRNAs dropping out would be deemed essential and the selection approach to identify genes important for resistance to clodronate liposomes. The latter function is explored as clodronate liposomes are widely used across species and experimental systems to deplete macrophages and phagocytic cells like mosquito hemocytes. Following the genome wide screen, several genes conferring resistance to clodronate liposomes were examined in vivo. The manuscript uses a powerful genome-wide approach to dissect two discrete processes which together or separately could ultimately inform future vector control strategies. The work is significant given its novelty and the information gained, though there are numerous issues with the conclusions drawn and approaches used that, in this reviewer's opinion, need to be addressed.

-Were the drop out experiments or the selection experiments replicated in any way? And if not, how was transfection efficiency controlled for? Given there is no sequence collected between the plasmid library pre-transfection and the endpoint this assumes all members of the plasmid library were efficiency transfected it could be hard to distinguish a true drop out. In other words, how can a true drop out be distinguished from a drop out that is the result of never being successfully transfected. Experimental replication also increases the statistical power for determining essentiality.

-The manuscript title requires editing—the definition of essential used throughout the paper is a relatively loose one as what is being characterized is actually a decrease in fitness (growth rate) in a competitive environment that results in drop off rather than true essentiality. For the study system in question an essential gene would be one in without which the cell could not grow. Having a negative impact on growth or fitness is not synonymous with essentiality. Further, it is never clearly stated that these are 'essential' genes under a given, in this case, basal, growth condition and in a given cell type/line.

-In the paragraph starting at line 507 how are expressed genes defined? And how many of the original 'essential' genes list were deemed false-discoveries due to a lack of expression and how do expression levels of the 'essential' gene set compare with the non-essential genes under the basal growth conditions. Finally the expression data derive from a study that used perfused hemocytes- how comparable are these with the cell line used in the current work.

-Of the 10 potential targets for RNAi, how do the 5 for which RNAi approaches failed to work compare in their level of expression under basal conditions? How does their expression differ between hemocytes and whole mosquitoes or other mosquito tissue?

The authors should provide more information on their depth of sequence and the resulting power to distinguish true drop out from low frequency.

-Other than a short exploration of serpent as a cell growth regulator, there is no in vivo validation of the essential gene set. Cell viability or growth assays are a viable way to assess a subset of the large number of essential genes ideally with a focus on those of differing degrees of essentiality through use of expression data.

-The authors focus on shared essential genes between Anopheles and Drosophila as well as humans, and do not explore at all the essential genes identified that may be specific to mosquitoes. Given the novelty of the system as an insect vector of public health importance a understanding of essential genes specific/novel to mosquitos bears exploration and discussion.

Throughout the figures, both in text and supplementary, sample sizes and biological replicate number are needed. These are needed to allow a judgement on robustness and rigor.

- Supplementary Figure 2 how many replicate P. berghei infection were done? And does the analysis couple infection prevalence with infection intensity? Without experimental replication
- How were intergenic regions control? To include or exclude predicted regions of regulatory function/lncRNAs/eRNAs etc.
- Were mosquito fitness and survival monitored during the RNAi mediated depletion of gene expression experiments?
- In Figures 2B, 5C and 5D, what is the meaning of the bar color—it is never described.

Reviewer #2

(Remarks to the Author)

In the present study, Mameli et al. describe a genome-wide CRISPR knockout screening platform developed for Anopheles mosquito cells. The study identified 1280 essential genes involved in fundamental cell processes and another set of genes conferring resistance to clodronate liposomes, which are used to ablate immune cells.

It is the first comprehensive study describing a forward genetic CRISPR screen in mosquito cells and the study is skillfully planned and executed. Following the screens, target candidate genes were meticulously validated in vivo. Using a combination of drugs and gene knockdowns, molecular factors involved in clodronate liposome function were identified. The conclusions are supported by the results, and I listed below a series of comments and questions, which hopefully will help improve the manuscript further.

Major comments :

1) Line 51: The choice of the Sua-5b cell line model could be justified further. Have other cell lines been tested ? Why has it been chosen for the screen ? Has the genome of the cell line been sequenced, and how does it compare to the mosquito genome ? And finally, which percentage of the transcriptome is detectable in this cell line – in other terms, what are the limits inherent to using this cell line ?

2) Line 104: How was the time of 8 weeks of outgrowth chosen ? Have preliminary studies been performed at earlier or later time points, and if so, how comparable are the obtained results ?

3) Fig 2a : To assess IC50, authors have estimated ATP levels, which is not the most standard way to assess apoptosis (the actual effect of clodronate treatment). Would the results have differed if the readout was apoptosis directly ? And if so, how would it have affected the results of the screen ?

4) Line 229 : Regarding the temporal data regarding the liposome uptake, have the authors assessed the duration of the depletion effect in vivo ? This information would be very useful to know.

Minor comments :

5) Line 86: The term “multiple” is rather vague, and it would be better to be precise. How many screens have in fact been performed in this study ?

6) Line 95: Would it be possible to provide in supplementary data the list of genes that were not targeted in the screen – again, highlighting the limits of the study and the specific cell line it exploited ? Also, it would be nice to discuss the potential use of primary mosquito cells for screens, as has been done for Drosophila.

7) Line 493: I think there might be a mistake regarding the size of the dishes, which is probably 150 mm.

8) Fig 2a: How do the authors explain the peak of ATP levels in the case of both treatment, just before the effects of the treatment are starting to be visible ?

9) Figure 2, panel C: It would be good to highlight the overlapping genes between the 3 plots by giving them another color.

Version 1:

Reviewer comments:

Reviewer #1

(Remarks to the Author)

The revised manuscript thoroughly addressed my previous concerns, and I am in support of publication.

Reviewer #2

(Remarks to the Author)

I thank the authors for addressing my comments, I believe the manuscript is well improved and ready for publication.

Response to Reviewer's comments

NCOMMS-24-68316

"A genome-wide CRISPR screen in *Anopheles* mosquito cells identifies essential and immune cell function-related genes"

A detailed response to each of the reviewer comments is listed below. All changes in the manuscript text made in response to the reviewer's comments are denoted by the highlighted text in our attached "Related Manuscript File".

Reviewer #1

This manuscript builds on a proof of principle paper published by the Perrimon group in 2022 <https://doi.org/10.1038/s41467-021-27129-3> which developed computational and biological tools and demonstrated that a genome wide CRISPR screen was possible in a mosquito hemocyte-like cell line. Here the authors apply a genome wide CRISPR screen in the *Anopheles coluzzii* hemocyte-like Sua-5b line for which they had previously generated tools to identify essential genes and those associated with immune cell function, specifically resistance to clodronate liposomes. The authors employ a drop out approach to identify essential genes such that those sgRNAs dropping out would be deemed essential and the selection approach to identify genes important for resistance to clodronate liposomes. The latter function is explored as clodronate liposomes are widely used across species and experimental systems to deplete macrophages and phagocytic cells like mosquito hemocytes. Following the genome wide screen, several genes conferring resistance to clodronate liposomes were examined in vivo. The manuscript uses a powerful genome-wide approach to dissect two discrete processes which together or separately could ultimately inform future vector control strategies. The work is significant given its novelty and the information gained, though there are numerous issues with the conclusions drawn and approaches used that, in this reviewer's opinion, need to be addressed.

We would like to thank the reviewer for their comments. We believe that we have addressed all of your comments in our revised manuscript. Our responses to each individual comment are listed below.

Reviewer Comments

- 1. Were the drop out experiments or the selection experiments replicated in any way? And if not, how was transfection efficiency controlled for? Given there is no sequence collected between the plasmid library pre-transfection and the endpoint this assumes all members of the plasmid library were efficiency transfected it could be hard to distinguish a true drop out. In other words, how can a true drop out be distinguished from a drop out that is the result of never being successfully transfected. Experimental replication also increases the statistical power for determining essentiality.**

Regarding replicates: We appreciate the reviewer's comments and agree there is value in replication. We had performed two biological replicates for the fitness screen but reported results for only one because the outcome was better, likely due to better transfection efficiency, resulting in a large list of genes at high statistical significance. However, we fully agree that it is appropriate for us to provide data from both biological replicates of the fitness screen. In the revised manuscript, we now report data for both replicate 1 and replicate 2. We now discuss replicate 2 and compare replicates 1 and 2 in the main text, and added a comparison of the replicates to Fig. 1 (new Fig. 1c and Fig. 1d). We also now include data from both replicates in Supplemental Table 2 including a comparison of Z-scores (all data in the MLE analysis tab, and tabs added for each dataset alone and the list of gene identified in both screens at FDR=0.05 or 0.10). In addition, we show a comparison of beta-score values for all genes for both replicates in a new figure, **Supplemental Fig. 1**.

Regarding the likelihood that a gene would score as a false positive because of low transfection efficiency: The experimental approach, which includes multiple sgRNAs per gene and MAGeCK MLE analysis, which is standard in the field, corrects for this possibility. Specifically, differences in abundance in the starting plasmid library, as detected by NGS, are incorporated in the MAGeCK MLE analysis when comparing to the cell pools. We further clarified in the Methods section that “The sgRNAs that had low representation in the plasmid library (readcounts less than 135) were trimmed prior to MLE analysis.” We also note that the number of genes that scored in replicate 2 of the fitness screen at the same FDR thresholds (0.05 or 0.10) was lower, not higher, compared with the number of genes that scored at the same statistical significance in replicate 1, and 87% (393/450) of hits identified in replicate 2 were also found in replicate 1.

- 2. The manuscript title requires editing—the definition of essential used throughout the paper is a relatively loose one as what is being characterized is actually a decrease in fitness (growth rate) in a competitive environment that results in drop off rather than true essentiality. For the study system in question an essential gene would be one in without which the cell could not grow. Having a negative impact on growth or fitness is not synonymous with essentiality. Further, it is never clearly stated that these are ‘essential’ genes under a given, in this case, basal, growth condition and in a given cell type/line.**

We agree that the term “fitness genes” is more inclusive and appropriate than “essential genes” when describing the screen data. We have revised the title and the wording throughout the manuscript accordingly. We maintain use of “essential” only when appropriate, e.g., in discussing essential cellular components or the “human core essential genes” list, which was reported by another group using that terminology.

- 3. In the paragraph starting at line 507 how are expressed genes defined? And how many of the original ‘essential’ genes list were deemed false-discoveries due to a lack of expression and how do expression levels of the ‘essential’ gene set compare with the non-essential genes under the basal growth conditions. Finally, the expression data derive from a study that used perfused hemocytes- how comparable are these with the cell line used in the current work.**

Expressed genes were defined based on log-transformed transcript abundance values ($\log_{10}[\text{TPM} + 0.01]$) with genes having expression values ≥ 1 (corresponding to $\text{TPM} \geq 1$) categorized as “expressed.” This classification is based on our re-analysis, reported in Viswanatha, Mameli, et al. (2021) in Nat Comm, of RNA-seq data for the Sua-5B cell line that was deposited at BioProject (Accession ID PRJNA238691) by the Broad Institute, using the AgamP4.12 annotation. We have clarified this in the Methods section and included this information in the legend to **Supplemental Table 2**.

The reviewer raises an important point that warrants clarification. Among the 1,280 genes identified as significant fitness candidates at a false discovery rate (FDR) threshold ≤ 0.05 in replicate 1, 27.6% were classified as non-expressed in our transcriptomic dataset ($\log_{10}[\text{TPM} + 0.01] < 1$). While this may appear contradictory, it is important to distinguish between the statistical methodology used in our screen and how expression data were used to estimate confidence. First, note that MAGeCK maximum likelihood estimation method (MLE) analysis is computed independently from gene expression data. Second, to estimate the false discovery rate (FDR), we subsequently leveraged RNA-seq expression data from the same cell line, based on the rationale that true fitness genes are generally expected to be transcriptionally active ($\text{TPM} \geq 1$). As described in the Methods, genes were ranked by Z-score, binned, and evaluated for the presence of non-expressed genes, which served as proxy for false positives. However, the FDR in this context is a statistical tool for setting a confidence threshold and does not imply that all genes below the expression cutoff are irrelevant for fitness. Due to the discrete nature of guide-level dropout effects and the variability in transcriptomic

measurements, especially regarding low-expressed genes, some genes classified as non-expressed may still produce reproducible phenotypes in the screen. Furthermore, the threshold used ($\text{TPM} \geq 1$) is based on a single bulk RNA-seq dataset and may not reflect low-level, transient, or cell-state-specific expression patterns that are relevant for fitness.

Importantly, as shown in the accompanying plot (showing the distribution of expression values across fitness genes), the majority of genes classified as “non-expressed” yet scoring as fitness hits show expression values near the cutoff (e.g., $\text{TPM} > 0.5$), supporting the idea that these genes are not strictly silent. Therefore, the inclusion of some non-expressed genes among fitness hits does not necessarily indicate erroneous results but rather reflects the sensitivity of pooled CRISPR screening to detect functional dependencies beyond steady-state expression levels. We now incorporate this information in the main text and provide summary tables and representative plots in a separate tab (“fitness score Vs expression”) in **Supplementary Table 3**. The methods section was updated accordingly.

To further address the relationship between gene fitness and transcript abundance, as suggested by the reviewer, we compared expression levels of fitness genes ($\text{FDR} \leq 0.05$) to all non-fitness genes analyzed in our fitness screen. Our analysis revealed a pronounced shift towards higher expression among fitness genes: Fitness genes ($n = 1,280$) had a mean $\log_{10}(\text{TPM} + 0.01)$ of 1.44 and a median of 1.37, with an interquartile range (IQR) of 0.97 to 1.81. In contrast, non-fitness genes ($n = 8,125$) had a markedly lower mean of -0.12 and median of 0.05, with an IQR spanning -2.00 to 1.04. These distributions suggest that the vast majority of fitness genes are actively expressed, with many falling within the top quantiles of transcript abundance. As shown in the violin plot, fitness genes form a dense cluster among the most highly expressed genes, mainly enriched for ribosomal components, which is consistent with the expectation that essential cellular functions are maintained at high expression levels. This clear enrichment supports the robustness of our screen and further validates the observed fitness phenotypes. We now incorporate this information in the main text and provide summary tables and representative plots in a separate tab (“fitness score Vs expression”) in **Supplementary Table 3**. The methods section was updated accordingly.

4. **Of the 10 potential targets for RNAi, how do the 5 for which RNAi approaches failed to work compare in their level of expression under basal conditions? How does their expression differ between hemocytes and whole mosquitoes or other mosquito tissue? The authors should provide more information on their depth of sequence and the resulting power to distinguish true drop out from low frequency.**

We appreciate the reviewer's comment as a possible explanation to our varying levels of success in performing RNAi on the 10 candidate genes identified in the clodronate screen. In our revised manuscript, we provide additional data examining gene expression and hemocyte-enrichment of the candidate genes using existing data sets (PMIDs: 19940242, 32855340, 34318744) to their potential impacts on the efficiency of RNAi. Summarized in **Supplementary Fig. 8**, our analysis suggests that there are no obvious patterns that explain why some genes are more amenable to gene-silencing *in vivo*.

Regarding the second part of the question (more information on depth of sequence etc.), we interpret that the reviewer is asking about the cell screen. We used 20 million reads per sample, which is within the standard range in the field. Transfection efficiency does seem to have impact on the total number of hits above a given threshold. Please refer to our in-depth response to question 1.

5. **Other than a short exploration of serpent as a cell growth regulator, there is no *in vivo* validation of the essential gene set. Cell viability or growth assays are a viable way to assess a subset of the large number of essential genes ideally with a focus on those of differing degrees of essentiality through use of expression data.**

We agree that a more thorough validation of the identified candidate fitness genes is needed to better validate the gene list and understand the contributions of these genes to cell viability, growth, or other aspects of cell fitness. While these experiments would have value to enhance our understanding of mosquito biology and for potential translational approaches to target mosquito fitness in the future, we believe that the experimental validation of these hits would be a significant effort beyond the scope of this study. Moreover, we would argue that validation in mosquito cells would only be informative, e.g., to help define false discovery rates beyond what we are able to infer based on gene expression data, if we were to test tens or hundreds of candidates. Even in cell lines, that would be a big project. Furthermore, validating what might arguably be the most interesting cases, such as putative cell type-specific or species-specific fitness genes, would require extensive testing using multiple approaches, again beyond the scope of this study. Lastly, we did identify the experimentally validated fitness gene *Rho1* (now noted in the main text) and suggest that the results of gene set enrichment analysis (GSEA) are sufficient to assess the quality of the fitness gene datasets, i.e., we observe clear statistically relevant enrichment in expected categories using multiple reference sets for GSEA.

Regarding *in vivo* validation, at present, additional *in vivo* validation experiments in *An. gambiae* are relatively impractical given that the system is less genetically tractable than other insects such as *Drosophila*. RNAi experiments are performed in adult mosquitoes through the injection of dsRNA, where gene candidates can only be evaluated for adult fitness or viability measurements. Even if a knockdown is possible, any phenotypes may not capture the gene requirements at earlier stages of development. Moreover, while CRISPR-mediated knockouts are possible, these are inefficient, laborious, and expensive to perform for one or more candidates.

- 6. The authors focus on shared essential genes between Anopheles and Drosophila as well as humans, and do not explore at all the essential genes identified that may be specific to mosquitoes. Given the novelty of the system as an insect vector of public health importance an understanding of essential genes specific/novel to mosquitos bears exploration and discussion.**

We agree that it is likely that some mosquito-specific fitness genes are included in the dataset we report here and merit further exploration. However, it is important to note that both the *Anopheles* and *Drosophila* fitness datasets are based on cell lines and may reflect requirements of these cell lines rather than universal fitness genes across tissues or developmental stages for each species. To address the reviewer question we analyzed our high confidence gene set (1280 genes at FDR=0.05 from Replicate 1). Among these, 146 mosquito genes with *Drosophila* orthologs did not score as fitness genes in *Drosophila*. Additionally, 61 genes have no identifiable orthologs in *Drosophila*, 59 of which lack both human or fly orthologs based on DIOPT analysis.

To aid interpretation, we have added new tabs to **Supplementary Table 3**: one listing *Anopheles* genes that have *Drosophila* orthologs, but do not score as fitness genes in *Drosophila*; and a second tab that lists genes with no orthologs in *Drosophila* or humans. For the latter we include available functional annotation from Pfam and other protein domain-based resources as well as ortholog mapping to other representative mosquito species. We also updated the main text in results and discussion sections highlighting these findings.

- 7. Throughout the figures, both in text and supplementary, sample sizes and biological replicate number are needed. These are needed to allow a judgement on robustness and rigor.**

We appreciate the comment and want to be fully transparent with our data and analysis. We have added information about sample sizes and biological replicates where applicable throughout our revised manuscript and supplemental figures.

- 8. Supplementary Figure 2 how many replicate P. berghei infection were done? And does the analysis couple infection prevalence with infection intensity? Without experimental replication**

In our revised manuscript, we provide additional experimental details and infection prevalence data. These changes are present in the revised figure and figure legend (now **Supplementary Fig. 3**).

- 9. How were intergenic regions control? To include or exclude predicted regions of regulatory function/lncRNAs/eRNAs etc.**

Intergenic control sgRNAs were selected in regions located at least 10 kb away from any annotated genes, including protein-coding genes, lncRNAs, and other known transcript types. However, we cannot fully exclude the possibility that some of these sgRNAs may fall within unannotated regulatory regions (e.g., long-distance enhancers), as such features are not well-annotated in mosquitoes. We included 400 intergenic control sgRNAs based on the reasonable assumption that the majority do not overlap functional regulatory elements.

Importantly, our analyses relied on the overall distribution of these 400 control sgRNAs, rather than on individual sgRNAs, to assess background signal. We now include this information in **Supplementary Table 1** as part of our revised manuscript.

10. Were mosquito fitness and survival monitored during the RNAi mediated depletion of gene expression experiments?

Since RNAi of the candidate genes showed no obvious effects on the survival of naïve (sugar-fed) mosquitoes, we did not directly measure fitness or survival, as further investigation seemed unnecessary. However, we cannot rule out the potential that the silencing of these candidate genes may influence survival following blood-feeding or infection (*Plasmodium*, bacteria, etc.), or that these genes may have an effect on reproductive fitness. Given the amount of work involved in performing these experiments under several physiological conditions, and for little expected gain, we opted to not include these additional experiments.

Similar to our response to “Comment #5” above, our RNAi experiments are performed in adult mosquitoes, a timepoint that may not fully address if the knockdown of a gene candidate has any contributions to mosquito fitness or survival. While it is possible to generate CRISPR knockouts in mosquitoes, this remains a challenging, laborious, and time-consuming process to evaluate multiple gene candidates, especially in *Anopheles gambiae*, which is the least genetically tractable mosquito system.

11. In Figures 2B, 5C and 5D, what is the meaning of the bar color—it is never described.

We apologize for the oversight. The bar colors are intended to help highlight those genes that display significant differences as compared to controls. We have added text to the figure legends of these respective figures to better define their visual purpose.

Reviewer #2

In the present study, Mameli et al. describe a genome-wide CRISPR knockout screening platform developed for *Anopheles* mosquito cells. The study identified 1280 essential genes involved in fundamental cell processes and another set of genes conferring resistance to clodronate liposomes, which are used to ablate immune cells. It is the first comprehensive study describing a forward genetic CRISPR screen in mosquito cells and the study is skillfully planned and executed. Following the screens, target candidate genes were meticulously validated in vivo. Using a combination of drugs and gene knockdowns, molecular factors involved in clodronate liposome function were identified. The conclusions are supported by the results, and I listed below a series of comments and questions, which hopefully will help improve the manuscript further.

We would like to thank the reviewer for their comments and intent to help us improve our manuscript. We believe that we have addressed all of your comments in our revised manuscript and the individual comments below.

Major comments

- 1. Line 51: The choice of the Sua-5b cell line model could be justified further. Have other cell lines been tested? Why has it been chosen for the screen? Has the genome of the cell line been sequenced, and how does it compare to the mosquito genome? And finally, which percentage of the transcriptome is detectable in this cell line – in other terms, what are the limits inherent to using this cell line?**

We used the Sua-5B cell line since it is one of a small number of established *Anopheles* cell lines that are readily available and amenable to methods compatible with setting up the screening system (e.g., single-cell cloning is feasible) and performing screens (e.g., it is transfectable and has a relatively high rate of proliferation). Moreover, this cell line has been characterized through previous RNA-seq experiments, providing additional expression data available for downstream analysis as noted above, which are now clarified in the Methods and in the Supplemental Table 2 legend of our revised manuscript.

The genome of the Sua-5B cell line is highly similar at the sequence level to the AgamP4.12 reference genome used for our sgRNA design. As described in the Methods, we conducted whole-genome sequencing of the Sua-5B cell line to optimize gRNA design in our previous study (Viswanatha, Mameli, *et al.*, *Nature Communications* 2021). sgRNAs based on the *Anopheles gambiae* reference genome (AgamP4.12) were compared to the Sua-5B genome, and only those matching the cell line were retained. Although we did not assemble a complete genome for Sua-5B, we aligned the sequencing data to both *Anopheles gambiae* and *Anopheles coluzzii* reference genomes to perform variant analysis. The Sua-5B cell line showed greater similarity to *Anopheles coluzzii*, with fewer SNPs compared to *Anopheles gambiae* (AgamP4.12 variants = 4,983,818; AcolM1.8 variants = 4,439,680; Δ = 544,138). This result was confirmed by a diagnostic PCR assay distinguishing these two closely related sister species and highlights the high sequence similarity between them, consistent with their recent divergence.

At our expression threshold (TPM \geq 1), 31.7% (4268/13460) of annotated genes (AgamP4.12) are expressed. This is somewhat lower than what we have observed previously in *Drosophila* S2R+ cell lines with approximately 45% genes expressed or for most human cell lines where 50-70% of genes are expressed.

The Sua-5B cell line is a heterogeneous cell line commonly recognized as “hemocyte-like”, was obtained and characterized by Giannoni *et al* (PMID: 11016929). To assess whether the Sua-5B cell line has similar transcriptional profile to hemocytes we compared the expression profiles of the cell line either to bulk RNA-seq data from perfused hemocytes (PMID: 32855340) or to hemocyte marker clusters identified in two different works from single cell RNA-seq experiments (PMID: 32855340, PMID: 34318744). From our mapping of Sua-5B gene expression to hemocyte-specific datasets and statistical validation, we confirm that this cell line compares positively with hemocytes either when comparing bulk RNA-seq data from hemocytes (~ 40% overlap, see new **Supplementary Table 5**) and also when we examine the expression of marker genes identified in single cell seq experiments, having up to ~90% overlap with single clusters. We now added **Supplementary Table 5** to include these data and analysis, updating the main text and Methods section accordingly in our revised manuscript.

While this finding does not alone define the identity of the cell line, it strongly suggests that the Sua-5B cell line exhibits an overall hemocyte-like transcriptional profile. Moreover, integrating transcriptional mapping with gene fitness information provides new insights into potential targets for future immunity and hemocyte related studies. We thank both reviewers for their comments regarding the cell line, which have helped to strengthen the scope and impact of the findings presented in this manuscript.

- 2. Line 104: How was the time of 8 weeks of outgrowth chosen? Have preliminary studies been performed at earlier or later time points, and if so, how comparable are the obtained results?**

The outgrowth time of eight weeks was chosen based on previous screens we performed with other drugs in *Anopheles* and *Drosophila* (e.g., in Viswanatha et al. 2018 eLife). This span of time was retained and proved sufficient to obtain a sizeable drug-driven enrichment when using drug concentrations at or above IC_{50} . We did not sequence samples at earlier or later time points, but we would expect higher enrichment if we treated the cells for additional time.

- 3. Fig 2a : To assess IC_{50} , authors have estimated ATP levels, which is not the most standard way to assess apoptosis (the actual effect of clodronate treatment). Would the results have differed if the readout was apoptosis directly? And if so, how would it have affected the results of the screen?**

In this study, IC_{50} is used as a technical parameter to define the drug concentration that reduces cell proliferation by 50%, regardless of whether cells die, their mode of death, or simply grow more slowly. This ensures a measurable fitness defect that enables enrichment of resistant clones within a practical timeframe. While apoptosis is likely the primary mechanism of clodronate toxicity, as suggested by prior literature, enrichment in a positive-selection CRISPR screen depends on cell growth rate rather than the specific mode of cell death. Measuring apoptosis directly might yield a different threshold (e.g., the concentration inducing caspase activation in 50% of cells), but for a survival-based screen, the relevant parameter is the concentration that slows proliferation. In this context, ATP levels serve as a practical proxy for cell viability and growth, making ATP-based IC_{50} an appropriate choice for the screen. That said, we agree that a change in drug concentration used as well as changes in the duration of treatment would significantly influence the dynamics and outcomes of enrichment.

- 4. Line 229: Regarding the temporal data regarding the liposome uptake, have the authors assessed the duration of the depletion effect in vivo? This information would be very useful to know.**

We now provide additional data as part of **Supplementary Fig. 5** in which we examine granulocyte depletion over the duration of ten days (every three days). Through these experiments, we demonstrate that clodronate liposomes deplete granulocyte populations across the entire time frame examined and that these cell populations do not appear to be replenished.

Minor comments

- 5. Line 86: The term “multiple” is rather vague, and it would be better to be precise. How many screens have in fact been performed in this study?**

The sentence “Herein, we perform multiple genome-wide CRISPR screens in an *Anopheles* mosquito cell line, identifying ~1300 essential genes responsible for cell viability and growth, as well as genes involved in the uptake and processing of clodronate liposomes.” Now reads: “Herein, we perform two types of genome-wide CRISPR screen ...”. For fitness genes, we performed two replicates using the same ‘drop-out’ assay. For clodronate, we performed two replicates for each of two different selection assay protocols.

- 6. Line 95: Would it be possible to provide in supplementary data the list of genes that were not targeted in the screen – again, highlighting the limits of the study and the specific cell line it exploited? Also, it would be nice to discuss the potential use of primary mosquito cells for screens, as has been done for *Drosophila*.**

We now added **Supplementary Table 6**, reporting statistics and list of *Anopheles* genes targeted/not targeted by the library in the various screens based on the latest gene annotation (AgamP4.14).

While it may be possible to one day perform screens on primary cells (most notably on immune cell/hemocyte populations), at present, the techniques for mosquito primary cell culture are limited or lacking. While there is evidence of *ex vivo* culture of midgut and fat body tissues for probing physiological questions shortly after dissection, we have yet to develop techniques for the culture of immune cells/hemocytes. As a result, there are significant technical challenges for the development of primary cell culture in mosquitoes similar to that of *Drosophila*.

7. Line 493: I think there might be a mistake regarding the size of the dishes, which is probably 150 mm.

You are correct and we apologize for the error. We now use mm instead of cm in our revised manuscript.

8. Fig 2a: How do the authors explain the peak of ATP levels in the case of both treatments, just before the effects of the treatment are starting to be visible?

We hypothesize that this phenomenon results from increased production of ATP at toxic drug concentrations, when a substantial portion of the cell population is affected, likely due to the activation of cell death mechanisms. We have observed this response occasionally also with other drug treatments that impact cell viability, and, as shown in Fig. 2a, it also occurs when very high concentrations of control liposomes are used. Although reports vary regarding the relationship between ATP levels and apoptosis, it has been shown (PMID: 15905877) that apoptosis is a programmed form of cell death that depends on ATP and both the total cellular ATP content and cytosolic ATP levels increase during the activation and execution phases of apoptosis. Overall, while this spike is an interesting observation, we believe it does not affect the calculation of the IC_{50} .

9. Figure 2, panel C: It would be good to highlight the overlapping genes between the 3 plots by giving them another color.

Given that, in the right panel, all genes are overlapping, since the plot is intended to show the relative enrichment (RRA rank) of all genes in the two treatments, we could not assign a single color, as they would all visually overlap. We have clarified the legend in Figure 2 to explain that we prioritized visualization of the top hits in each screen: the top 8 hits are shown in black in the left and center panels. Additionally, genes highlighted in magenta represent all hits (including some within the top 8) that were selected for follow-up studies, as they will be the focus of further investigation.